

# Forward Model Emulator for Atmospheric Radiative Transfer Using Gaussian Processes And Cross Validation

Otto Lamminpää[1], Jouni Susiluoto[1], Jonathan Hobbs[1], James McDuffie[1], Amy Braverman[1], and Houman Owhadi[2]

[1]Jet Propulsion Laboratory, California Institute of Technology, Pasadena, CA, USA
[2]California Institute of Technology, Pasadena, CA, USA

**Correspondence:** Otto Lamminpää (otto.m.lamminpaa@jpl.nasa.gov)

**Abstract.** Remote sensing of atmospheric carbon dioxide ($CO_2$) carried out by NASA's Orbiting Carbon Observatory-2 (OCO-2) satellite mission and the related Uncertainty Quantification (UQ) effort involves repeated evaluations of a state-of-the-art atmospheric physics model. The retrieval, or solving an inverse problem, requires substantial computational resources. In this work, we propose and implement a statistical emulator to speed up the computations in the OCO-2 physics model. Our approach is based on Gaussian Process (GP) Regression, leveraging recent research on Kernel Flows and Cross Validation to efficiently learn the kernel function in the GP. We demonstrate our method by replicating the behavior of OCO-2 forward model within measurement error precision, and further show that in simulated cases, our method reproduces the $CO_2$ retrieval performance of OCO-2 setup with orders of magnitude faster computational time. The underlying emulation problem is challenging because it is high dimensional. It is related to operator learning in the sense that the function to be approximated is mapping high-dimensional vectors to high-dimensional vectors. Our proposed approach is not only fast but also highly accurate (its relative error is less than 1%). In contrast with Artificial Neural Network (ANN) based methods, it is interpretable and its efficiency is based on learning a kernel in an engineered and expressive family of kernels.

## 1 Introduction

Climate change, one of the most significant global environmental challenges, is primarily attributed to anthropogenic carbon emissions, which have accelerated the increase of carbon dioxide ($CO_2$) in the atmosphere, posing a threat to Earth's future. The industrial revolution marked the onset of increased $CO_2$ emissions due to the extensive use of fossil fuels in various industries, such as transportation, manufacturing, and agriculture. The Intergovernmental Panel on Climate Change underscores $CO_2$'s potent effect on planetary warming due to significant radiative forcing (IPCC, 2014). The atmospheric concentration of this trace gas is increasing at ever faster rate, and as of May 2023, the measured $CO_2$ at Mauna Loa station was 424.0 ppm, a 3 ppm increase from a year before (421.0 ppm in May 2022). Although the global terrestrial biosphere and oceans each take up about 25% of these emissions (Friedlingstein et al., 2022), this balance may not be sustainable, which might lead to unpredictable



feedbacks in the carbon cycle and the global climate system. These couplings between the Earth's climate system and the carbon cycle can introduce significant uncertainty in future climate change projections (Friedlingstein et al., 2014), which
further renders mitigation efforts increasingly challenging.

For reliable climate modeling and future scenario prediction, it's crucial to estimate carbon flux accurately (e.g. Carbon-Tracker, Peters et al. (2007)), which involves quantifying both the sources and natural sinks of carbon. However, current *in situ* measurement networks are primarily deployed in the northern midlatitudes, leaving areas like the tropics underrepresented. This lack of extensive coverage results in large uncertainties in flux estimates, underscoring the need for a more comprehensive
global measurement network.

To provide a significant increase in coverage and resolution to the ground based data set, global estimates of total column mole-fraction $CO_2$ , denoted $XCO_2$, are collected using satellite-borne spectrometers. These instruments include the Japanese Greenhouse gases Observing SATellite (GOSAT, Kuze et al. (2009)), operational since January 2009, follow-on GOSAT-2 (Imasu et al., 2023) launched in October 2018, the Orbiting Carbon Observatory-2 from NASA (OCO-2, Crisp et al. (2012)),
launched in July 2014, and the OCO-3 instrument (Eldering et al., 2019) taken to the International Space Station May 2019. Planned future missions include the Geostationary Carbon Cycle Observatory (GeoCarb, Moore III et al. (2018)), the European $CO_2$ Monitoring Mission (CO2M, Sierk et al. (2019)) and the Global Observing Satellite for Greenhouse gases and Water cycle (GOSAT-GW, Kasahara et al. (2020)). In this work, we focus exclusively on OCO-2, which, like all the above mentioned missions, measures solar radiance at the top of the atmosphere, reflected by Earth's surface and attenuated by atmospheric
scattering and absorption by trace gases and aerosols. From these observed radiances, the OCO-2 mission uses a framework called *Optimal Estimation* (OE, Rodgers (2004)) to solve the related Bayesian inverse problem (see e.g. Kaipio and Somersalo (2005)), referred to as a retrieval. OE is an iterative algorithm, returning an estimate of posterior mean and covariance as a Gaussian approximation to the non-linear retrieval problem. Operationally the retrieval problem is solved using the Atmospheric Carbon Observations from Space (ACOS) software (O'Dell et al., 2018), which implements OE using a state-of-the-art
atmospheric Full Physics model (FP). Processing OCO-2 measurements with ACOS algorithm is a computationally intensive task, and not all soundings are currently processed for this reason. Computational speed is also a major hindrance for reprocessing the measurement record with improved algorithms and thus a limiting factor in releasing the improved data to the user community. These issues are certain to get even worse with upcoming wider swath missions like CO2M and GOSAT-GW, as evidenced by another greenhouse gas imaging mission, Tropospheric Ozone Monitoring Instrument (TROPOMI, Veefkind
et al. (2012)) from regularly reprocessing their data record, which is more than 20 times greater in size than that of OCO-2.

As with all inverse problems, some approximations and assumptions have to be made in the ACOS algorithm. The resulting $XCO_2$ estimates have to be validated and bias corrected using ground-based measurements from the Total Carbon Column Observing Network (TCCON, Wunch et al. (2017)), and COllaborative Carbon Column Observing Network (COCCON, Frey et al. (2019)) as a reference. These sites are concentrated on the Northern Midlatitudes, and as a result of this coverage issue
and the imperfections in the FP model, significant systematic errors persist in the data set. (See e.g. Kiel et al. (2019) for effect of systematic errors, and Cressie (2018) for overview of statistical treatment of, and issues in the retrieval). Considerable effort has been exerted to tackle the high accuracy (less than 0.3 parts per million (ppm) in scenes with background levels of





around 410 ppm), and high precision (standard errors less than 0.5 ppm) requirements of ingesting OCO-2 into flux inversion, which is the primary application of the data product. (Gurney et al. (2002), Patra et al. (2007), Liu et al. (2017), Palmer et al. (2019), Crowell et al. (2019), Peiro et al. (2022), Byrne et al. (2022)). Recent advancements of applying Markov Chain Monte Carlo (MCMC, Brynjarsdóttir et al. (2018), Lamminpää et al. (2019)) for non-Gaussian posterior characterization, and Simulation Based Uncertainty Quantification (Turmon and Braverman (2019), Braverman et al. (2021)) for capturing the overall uncertainty in the retrieval pipeline have been successfully deployed to address persisting retrieval errors. These methods, although comprehensive, suffer equally from computational speed issues as they require extensive amounts of FP evaluations.

Computational speed issues in OE retrievals have been attempted to address in several ways. Neural network (NN) based machine learning approaches (David et al. (2021), Mishra and Molinaro (2021), Bréon et al. (2022)) have been implemented to a combination of real world radiance data and model atmospheres. The OCO-2 forward model itself was sped up by using a surrogate model (Hobbs et al., 2017) that only partially considered the physical processes present in the FP model, and more recently by using a Gaussian Process (GP) emulator (Ma et al., 2019) for replicating the output of the FP model. In this paper, we will take similar approach using GPs, but with several improvements and an application to solving the retrieval problem with the help of closed-form Jacobians required in the gradient-based algorithm. Our approach will leverage recent novel techniques for GP parameter learning called *Kernel Flows* (Owhadi and Yoo, 2019), and training data generation via evaluating the FP model using the Reusable Framework for Atmospheric Composition (ReFRACtor) (McDuffie et al., 2018). We will demonstrate the accuracy of forward model emulation against a held out test set of FP evaluations, and further demonstrate the ability of our emulator to replicate the OE retrieval performance of ReFRACtor FP model in a fraction of computational time. Our approach achieves a remarkably low prediction error, less than 1% ("within measurement error limits"), which is an excellent result in the field of more general *operator learning*. Strategies to achieve learning more complicated operators, like the FP in our case, often involve a NN based architecture (Lu et al. (2021), Li et al. (2022)). Our approach follows the example set by Batlle et al. (2023) that kernel methods are competitive in operator learning.

The rest of the paper is organized as follows. Section 2 will describe in detail the GP regression, kernel learning and resulting forward model emulator. Section 3 will further elaborate on the details of the OCO-2 retrieval algorithm, the state vector, and FP model describing atmospheric radiative transfer. Section 4 will detail the emulator implementation of ReFRACtor FP model and assess its performance. Section 5 will show results of our emulator used in a simulated $XCO_2$ retrieval context, and finally Section 6 will provide concluding remarks and ideas on future work and applications.

## 2 Gaussian Process Emulator

*Gaussian Process (GP) regression* (Rasmussen and Williams, 2006) (also called *kriging* in spatial context: Cressie (1993); Stein (1999)) is a well studied methodology for approximating any continous function to an arbitrary accuracy, leveraging training data and a *kernel function* prescribed a priori. In this section, we outline the basic theory of GP regression and outline





our approach to modeling the continous function between atmospheric state vectors $\boldsymbol{x}$ and radiances $\boldsymbol{y}$ observed by the OCO-2 instrument.

## 2.1   Gaussian Process Regression

To construct an emulator for the forward model $F(\boldsymbol{x})$, we employ Gaussian Process (GP) regression to predict a *label* $z^* \in \mathbb{R}$ at a new *state* $\boldsymbol{x}^* \in \mathbb{R}^m$. A GP is defined by a *kernel function* $k(x, x') : \mathcal{X} \times \mathcal{X} \to \mathbb{R}$, where in the cases studied in this work
$\mathcal{X} = \mathbb{R}^m$. We denote by $\boldsymbol{\Gamma}(\boldsymbol{X}, \boldsymbol{X})$ the matrix of all kernel function evaluations over the training data $\boldsymbol{X} \in \mathbb{R}^{m \times N}$ of $N$ points with the entries $\boldsymbol{\Gamma}(\boldsymbol{X}, \boldsymbol{X})_{i,j} = k(\boldsymbol{x}_i, \boldsymbol{x}_j)$ where $\boldsymbol{x}_i, \boldsymbol{x}_j$ are the $i$th and $j$th training data points, respectively. Furthermore, $\boldsymbol{\Gamma}(\boldsymbol{x}^*, \boldsymbol{X})$ denotes the vector of kernel evaluations of state $\boldsymbol{x}^*$ against all training points $\boldsymbol{X}$. Using the training data together with vector of corresponding labels $\boldsymbol{z} \in \mathbb{R}^N$, a GP prediction of label (or function value) at a new state $\boldsymbol{x}^*$ is given by

$$z^* \equiv GP(\boldsymbol{x}^*) = \boldsymbol{\Gamma}(\boldsymbol{x}^*, \boldsymbol{X})(\boldsymbol{\Gamma}(\boldsymbol{X}, \boldsymbol{X}) + \sigma \mathbb{I})^{-1} \boldsymbol{z}, \tag{1}$$

where we have assumed w.l.g. that the training data are centered and thus the GP has a zero mean.

     **Remark**: In GP literature, the variance term $\sigma \mathbb{I}$ is usually taken to be the measurement error or local-scale unexplained variability in the training labels $\boldsymbol{z}$. However, since we are interested in reproducing the outputs of a computer code, the "measurements" are exact and hence there is no measurement error. It was shown in Owhadi and Yoo (2019) that learning the parameters of GP models from noiseless data is can lead to unstable predictive models and numerical singularities. For this
reason, we treat $\sigma$ as a regularization parameter, which captures the empirical mismatch between the model and the actual data, and optimize it together with other kernel parameters.

     In addition to point predictions, GP prediction can be associated with prediction uncertainty (the posterior variance of the GP), given by

$$\sigma^* = k(\boldsymbol{x}^*, \boldsymbol{x}^*) - \boldsymbol{\Gamma}(\boldsymbol{x}^*, \boldsymbol{X})(\boldsymbol{\Gamma}(\boldsymbol{X}, \boldsymbol{X}) + \sigma \mathbb{I})^{-1} \boldsymbol{\Gamma}(\boldsymbol{X}, \boldsymbol{x}^*). \tag{2}$$

The ability to preclude prediction uncertainties sets GP regression apart from many modern neural network based machine learning methods, which only provide a point estimate as a prediction. Large prediction variance can be an indication of departure from the support of training data set, indicating that GP is likely to lose its prediction skill. Additionally, uncertainty from the predictions can be propagated forward and accounted for in further applications of a GP based emulators.

## 2.2   Kernel Function

A crucial modeling choice in GP regression is specification of a kernel function. This task involves either expert knowledge of the domain structure, or some iterative trial and error search. In our application, we have empirically observed that a kernel function consisting of sum of Matérn and linear kernels yields excellent predictive performance. This is likely due to a locally near-linear behavior commonly assumed with the OCO-2 forward model being captured by the linear kernel, together with a largely flexible Matérn term that is known to capture a large variety of non-linear effects. Such kernel can also be differentiated





in closed form. The kernel function used throughout this work is given by

$$k(\boldsymbol{x}, \boldsymbol{x}') = \alpha_1 \left( 1 + \frac{\sqrt{3}}{l} \|(\boldsymbol{x} - \boldsymbol{x}')\|_{\mathcal{W}} \right) \exp \left( -\frac{\sqrt{3}}{l} \|(\boldsymbol{x} - \boldsymbol{x}')\|_{\mathcal{W}} \right) + \alpha_2 (\mathcal{W}\boldsymbol{x})^T (\mathcal{W}\boldsymbol{x}') \qquad (3)$$

where $\|(\boldsymbol{x} - \boldsymbol{x}')\|_{\mathcal{W}} = \sqrt{(\boldsymbol{x} - \boldsymbol{x}')^T \mathcal{W}^2 (\boldsymbol{x} - \boldsymbol{x}')}$, $\mathcal{W} = diag(\boldsymbol{w})$ is a diagonal matrix where $\boldsymbol{w} \in \mathbb{R}^m$ is a vector of weights, $l \in \mathbb{R}$ is a length scale parameter, and $\alpha_1, \alpha_2 \in \mathbb{R}_+$ are positive weights that are restricted to sum to 1.

Our interest will be in replicating the results of a gradient based optimization problem. Hence, in addition to fast evaluations of $F(\boldsymbol{x})$, we would also benefit from fast derivatives obtained from closed form expressions. Combining equations (1) and (3), we get

$$\frac{d}{d\boldsymbol{x}^*} z^* = \frac{d}{d\boldsymbol{x}^*} \boldsymbol{\Gamma}(\boldsymbol{x}^*, \boldsymbol{X})(\boldsymbol{\Gamma}(\boldsymbol{X}, \boldsymbol{X}) + \sigma \mathbb{I})^{-1} \boldsymbol{z}, \qquad (4)$$

where the derivative of the kernel function $\frac{d}{d\boldsymbol{x}^*} \boldsymbol{\Gamma}(\boldsymbol{x}^*, \boldsymbol{X})$ can be computed in closed form from eq. (3) using known matrix identities. The derivation of a closed form expression can be found in Appendix A.

## 2.3 Parameter Learning

Prediction quality of GP regression depends on identifying the hyperparameters $\theta$ that best fit the training data. In our case, following the form of our kernel function, we have $\theta = [\boldsymbol{w}, l, \sigma, \alpha_1, \alpha_2]$. Hyperparameters are commonly learned via optimization, using maximum likelihood estimation (MLE). This amounts to minimizing

$$\mathcal{L}(\theta) = -\frac{1}{2} \log\left[\det \boldsymbol{\Gamma}(\theta)\right] - \frac{1}{2} \boldsymbol{z}^T \boldsymbol{\Gamma}(\theta)^{-1} \boldsymbol{z}, \qquad (5)$$

where $\boldsymbol{\Gamma}(\theta) = \boldsymbol{\Gamma}(\boldsymbol{X}, \boldsymbol{X})$ evaluated at parameter values $\theta$. Although this method is usually robust and performs well, GP applications with high dimensional inputs and large amount of training data are known to be challenging due to inverse matrix and log determinant calculations. Numerous approaches have been suggested to tackle this problem (e.g. local approximations (Vecchia, 1988), (Datta et al., 2016)). Inspired by the Kernel Flows approach (Owhadi and Yoo, 2019) where kernel parameters are learned by minimizing a relative reproducing kernel Hilbert space norm, we propose a cross-validation root mean square error based method to be used in this work. The upsides of our approach are the ability to select small mini-batches on each training iteration, allowing for faster computations while avoiding expensive log-determinant calculations and inverting the large covariance matrices required in MLE. We will later show that our proposed method converges reliably and yields excellent predictions.

We start by selecting a mini-batch $\boldsymbol{X}^{\text{Batch}}, \boldsymbol{z}^{\text{Batch}}$ of size $N_{\text{Batch}}$ by randomly sampling from training data. We define a leave-one-out cross-validation loss function with respect to $L^2$ error by first considering taking out one data point from the training data, and using the rest to predict it. This can be achieved by modifying the GP prediction formula from eq. (1) and leaving out the $i$th data point, given by

$$\widetilde{GP}(\theta, i) = \widetilde{\boldsymbol{\Gamma}}(\theta)_{:,i}^T \left( \widetilde{\boldsymbol{\Gamma}}(\theta)^{-1} - \frac{\widetilde{\boldsymbol{\Gamma}}(\theta)_{:,i}^{-1} \widetilde{\boldsymbol{\Gamma}}(\theta)_{:,i}^{-T}}{\widetilde{\boldsymbol{\Gamma}}(\theta)_{i,i}^{-1}} \right) \boldsymbol{z}^{\text{Batch}} \qquad (6)$$





where $\widetilde{\boldsymbol{\Gamma}}(\theta) = \boldsymbol{\Gamma}(\boldsymbol{X}^{\text{Batch}}, \boldsymbol{X}^{\text{Batch}})$ is the $N_{\text{Batch}} \times N_{\text{Batch}}$ covariance over the mini-batch evaluated at parameter values $\theta$. A rank-one downdate $\widetilde{\boldsymbol{\Gamma}}(\theta)^{-1} - \frac{\widetilde{\boldsymbol{\Gamma}}(\theta)_{:,i}^{-1}\widetilde{\boldsymbol{\Gamma}}(\theta)_{:,i}^{-T}}{\widetilde{\boldsymbol{\Gamma}}(\theta)_{i,i}^{-1}}$ is used to remove the effect of $i$th data point from the inverse covariance matrix $\widetilde{\boldsymbol{\Gamma}}(\theta)^{-1}$. (See Stewart (1998) and Zhu et al. (2022) for details). Here, the notation $\widetilde{\boldsymbol{\Gamma}}(\theta)_{:,i}$ means all rows of the $i$th column. We then define the final loss function by using formula (6) to predict $z_i$ (the $i$th training label removed from the mini-batch) as

$$\rho(\theta) = \sum_{i=k_1}^{k_p} \left( \widetilde{GP}(\theta, i) - z_i \right)^2 + \epsilon \|\theta_0 - \theta\|^k, \tag{7}$$

where $i \in [k_1 \ldots k_p] \subset [1 \ldots N_{\text{Batch}}]$ is a subset of $p \leq N_{\text{Batch}}$ indices denoting elements of the mini-batch selected for prediction, which can be chosen as e.g. the entire mini-batch, or the $p$ nearest neighbors of the center point of the mini-batch. The regularization term with $k$ norm $\|\cdot\|^k$, some penalty magnitude $\epsilon$, and mean $\theta_0$, is included to ensure that kernel amplitude parameter values don't grow uncontrollably. This is done since we have observed empirically that letting non-identifiable parameters grow during optimization can lead to the optimizer getting "stuck", whereas this problem is not observed when regularizing the loss function. One may for example set $\theta_0$ to be a vector of 1's.

We can now optimize the kernel parameters parameters iteratively by repeatedly selecting mini-batches and updating $\theta$ along the gradient of $\rho(\theta)$, which is obtained by automatic differentiation using Julia's *Zygote* package (Innes, 2018). We note that as the mini-batch is selected at random, this method can be viewed as stochastic gradient descent. For this reason, we use the adaptive moment estimation (ADAM, (Kingma and Ba, 2017)) optimizer for finding the optimal value. Use of a momentum based optimizer is further recommended in this application as we have observed that the cost function often has several local minima. The optimization procedure is summarized in Algorithm 1. The final parameter value can be selected to be the one corresponding to the smallest loss function value achieved during training.

---

**Algorithm 1** Kernel Parameter Learning

---

**Input:** Kernel function $k$, training data $(\boldsymbol{X}, \boldsymbol{z})$, batch size $N_{\text{Batch}}$, number of prediction points $p$, number of iterations $N_{\text{Iter}}$.

**Output:** Matrix of kernel parameters $\boldsymbol{\Theta}$ and vector of loss values $\boldsymbol{R}$

1: Initialize $\theta_1 \leftarrow 1, \boldsymbol{\Theta} \leftarrow \boldsymbol{0}, \boldsymbol{R} \leftarrow \boldsymbol{0}$

2: **for all** $i$ in $1 \ldots N_{\text{Iter}}$ **do**

3:     $\boldsymbol{X}^{\text{Batch}} \leftarrow \boldsymbol{X}[rand(N_{\text{Batch}})], \quad \boldsymbol{z}^{\text{Batch}} \leftarrow \boldsymbol{z}[rand(N_{\text{Batch}})]$          // *Randomly select a mini-batch* $\boldsymbol{X}^{\text{Batch}}, \boldsymbol{z}^{\text{Batch}}$

4:     $\boldsymbol{R}[i] \leftarrow \rho(\theta_i)$             // *Compute loss* $\rho(\theta_i)$ *from* (7)

5:     $\boldsymbol{\Theta}[i] \leftarrow \theta_{i+1}, \quad \theta_{i+1} \leftarrow \theta_i + ADAM(\frac{\partial}{\partial \theta}\rho(\theta_i))$     // *Compute gradient* $\frac{\partial}{\partial \theta}\rho(\theta_i)$ *and update parameters* $\theta_i$ *using ADAM*

6: **end for**

7: **return** $\boldsymbol{\Theta}, \boldsymbol{R}$

---

## 2.4 Training Data Generation

As we aim to reproduce the performance of a function represented as computer code, we take advantage of the freedom to use a space filling design for $\boldsymbol{x}$ in $\mathbb{R}^m$ for training data creation. We first span the unit cube $[0, 1]^m$ with a *Sobol sequence* (Sobol,



1967) of $N$ points. In practice we employ Julia's *Sobol.jl* (Johnson, 2020) package for this step. Then, using information about the minimum and maximum physically feasible value of each input dimension, we scale the unit cube to span the whole state space. We further evaluate the computational model $F(x)$ at each training point, obtaining states $\boldsymbol{X} \in \mathbb{R}^{N \times m}$ and model outputs $\boldsymbol{Y} \in \mathbb{R}^{N \times n}$.

## 3   The Orbiting Carbon Observatory 2

In this section, we describe OCO-2 and the related measurements, physics model, state vector, and retrieval algorithm. Further information on these topics can be found in e.g. Connor et al. (2008), O'Dell et al. (2012), Crisp et al. (2012), O'Dell et al. (2018), and in the Algorithm Theoretical Basis document (ATBD) Crisp et al. (2021).

### 3.1   The OCO-2 Instrument

OCO-2 is a NASA operated satellite mission dedicated to providing data products of global atmospheric carbon dioxide con-
centrations (Crisp et al., 2004). The satellite is pointed towards Earth as it measures solar light reflected by Earth's surface and atmosphere, recorded as radiances. The OCO-2 instrument itself is composed of three spectrometers that measure light reflected from Earth's surface in the infrared part of the spectrum in three separate wavelength bands. These bands are centered around 0.765, 1.61 and 2.06 $\mu$m and are called the $O_2$ A-band (O2), the Weak $CO_2$ band (WCO2) and the Strong $CO_2$ band (SCO2), respectively. Each observation consists of 1016 radiances on separate wavelengths from each band (for more informa-
tion, see e.g. Crisp et al. (2017), Rosenberg et al. (2017)). These measurements are then used to infer a state vector containing information on atmospheric properties like $CO_2$ concentration on 20 pressure levels, surface pressure, temperature and aerosol optical depth (AOD). The state vector also includes surface properties like albedo, and solar induced chlorophyl fluorescence (SIF). The primary scalar quantity of interest is the column-averaged $CO_2$ concentration ($XCO_2$).

### 3.2   Atmospheric Radiative Transfer

A key part to inferring $XCO_2$ from observed radiances is construction of a computational atmospheric radiative transfer model which describes how solar radiation is propagated, reflected and scattered by Earth's surface and atmosphere. Together with an instrument model, this computer code is known as the Full Physics (FP) model, referred to in this work as

$$\boldsymbol{y} = F(\boldsymbol{x}, \boldsymbol{b}), \tag{8}$$

where $\boldsymbol{y}$ is output of the FP model, a wavelength-by-wavelength radiance, $\boldsymbol{x}$ is a state vector containing atmospheric and
surface information, and $\boldsymbol{b}$ are model parameters held fixed during data processing. Part of the radiance comes from absorption of radiation by atmospheric molecules, given by

$$I(\lambda) = f_0(\lambda) \cos(\tau_0) \cdot R(\lambda, \tau, \tau_0, \varphi - \varphi_0) \exp\left(-g(\lambda)\right) \tag{9}$$

where $\lambda$ is wavelength, s.t. the $j$th wavelength corresponds to the $j$th entry of randiance $y$, $f_0(\lambda)$ is the solar flux at the top of the atmosphere, $R(\lambda, \tau, \tau_0, \varphi - \varphi_0)$ is the reflectance of the surface, $g(\lambda)$ is a integral over radiation path length that sums





over for all modeled absorbers, $\tau$ and $\varphi$ are the observation zenith and azimuth angles, and $\tau_0$ and $\varphi_0$ are the corresponding solar zenith and azimuth angles. Observation and solar angles have a significant effect on the observed and modeled radiances, which will be important later in this work.

After calculating the absorption with equation (9), equations further describing atmospheric scattering are employed to solve for *atmospheric radiative transfer* (RT), which describes the total effect of atmosphere and surface to the scattered photons.
The FP framework further includes an instrument model, which describes the effects of the observing system to the *top of the atmosphere radiances*. These effects include fluorescence, instrument doppler shift, spectral dispersion and convolution with the instrument line shape (ILS) function, reducing the resolution from the finer RT grid to the coarser observational grid. On an abstracted level, this corresponds mathematically to

$$I_{OBS}(\lambda) = C_1(\lambda) \int_{-\infty}^{+\infty} RT(\lambda')ILS(\lambda,\lambda')d\lambda' + C_2(\lambda), \qquad (10)$$

where $C_1(\lambda)$ and $C_2(\lambda)$ denote the instrument effects other than convolution that can be expressed as multiplication and addition. Generally speaking, the instrument effects depend on different physical properties that can vary between detector arrays, while the RT portion of the forward model is constant within the instrument. This observation motivates us to focus on emulating the outputs of the RT, referred to as *monochromatic radiances*, after which instrument functions can be applied appropriately after the fact. Looking forward to operational integration of our emulator, this will reduce the complexity of the
emulated system and arguably make our task easier.

### 3.3  OCO-2 State Vector

The state vector elements comprising $x$ for the FP model are summarized in Table 1. Notably, we have divided the table in two parts. The upper half lists the previously-mentioned atmospheric and surface state vector elements that affect the RT part only, and the rest having to do with the instrument effects are in the lower half. This collection includes scaling factors for empirical
orthogonal functions (EOFs) that capture unmodeled offsets in the observed radiances O'Dell et al. (2018).





| State vector element | # elements | O2 | WCO2 | SCO2 |
|---|---|---|---|---|
| CO2 concentration profile | 20 | | ✓ | ✓ |
| H2O Scaling factor | 1 | ✓ | ✓ | ✓ |
| Surface Pressure (Pascals) | 1 | ✓ | ✓ | ✓ |
| Temperature Offset (Kelvin) | 1 | ✓ | ✓ | ✓ |
| Aerosol height, width and AOD | 12 | ✓ | ✓ | ✓ |
| O2 band albedo | 2 | ✓ | | |
| WCO2 band albedo | 2 | | ✓ | |
| SCO2 band albedo | 2 | | | ✓ |
| O2 band dispersion | 2 | ✓ | | |
| WCO2 band dispersion | 2 | | ✓ | |
| SCO2 band dispersion | 2 | | | ✓ |
| O2 band EOF scaling | 3 | ✓ | | |
| WCO2 band EOF scaling | 3 | | ✓ | |
| SCO2 band EOF scaling | 3 | | | ✓ |
| SIF parameters | 2 | ✓ | | |

**Table 1.** Elements of the OCO-2 state vector by functional group. The second column indicates the total elements per group. The check marks in the remaining columns indicate which wavelength bands are sensitive to changes on each variable.

In addition to state vector elements, the FP model is parametrized by a set of parameters that are held fixed based on auxiliary information, such as laboratory measurements or meteorological datasets. These parameters include instrument calibration details, spectroscopy properties for absorbing gases, land elevation, and aerosol microphysical parameters. These aerosol pa-

rameters arise from the selection of two dominant aerosol types as a function of space and time. All aerosol types have different optical properties. This choice is determined a priori by global maps based on meteorological knowledge and measurements (see Figure 1). The possible dominant aerosol types are dust (DU), sulphate (SO), sea salt (SS), organic carbon (OC), and black carbon (BC). While constructing the emulator, we will consider datasets with a fixed pair of dominant aerosol species in order to decouple their physical effects from the rest of state vector. Separate emulators can then be constructed for each pair

of aerosol species, and a selection of which one to use can be done by matching the measurement location with the appropriate types.





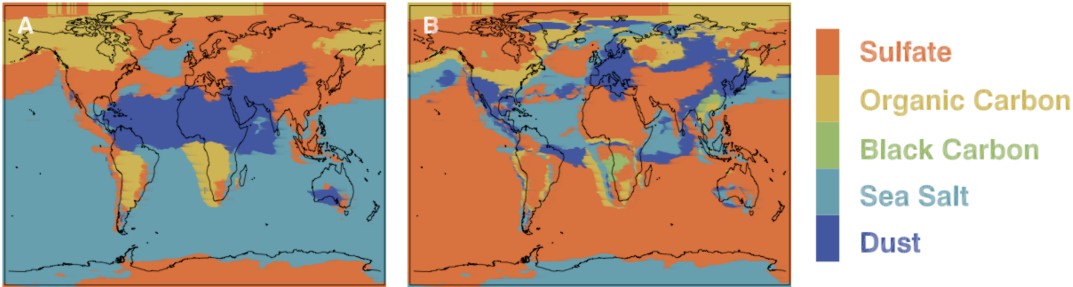

**Figure 1.** Example global map of A) primary and B) secondary aerosol types used. Image taken from (Boesch et al., 2015)

## 3.4 ReFRACtor

This work develops a proof-of-concept version of OCO-2 forward model emulator for a simulated case. For this reason and ease of access, we implement our simulations using The Reusable Framework for Atmospheric Composition (ReFRACtor, McDuffie

et al. (2018)). ReFRACtor is an extensible multi-instrument atmospheric composition retrieval framework that supports and facilitates combined use of radiance measurements from different instruments in the ultraviolet, visible, near-infrared and thermal-infrared. It has been open source since 2014 when it was first developed as the Level-2 processing code for OCO-2. Since 2017 the development team has worked to create a more general framework that supports more instruments and spectral regions. This framework has been developed to provide the broader Earth science community a freely licensed software package

that uses robust software engineering practices with well tested, community accepted algorithms and techniques. ReFRACtor is geared not only for the creation of end-to-end production science data systems, but also towards scientists who need a software package to help investigate specific Earth science atmospheric composition questions. Although ReFRACtor includes an implementation of a version of the OCO-2 production algorithm, the two have drifted since the initial inter-comparison comparison work was done. At that time it was validated against the B9.2.00 version of the software. For the most part mainly

bug fixes have been kept in sync between the two versions. Additionally the core radiative transfer algorithms are the same, which justifies the use of ReFRACtor for constructing our emulator at this stage. Some minor additional algorithmic features made their way into the ReFRACtor version of OCO-2 from the production version. For the most part the major discrepancy will be due to changes to configuration values not implemented in ReFRACtor. These include values such as a priori and covariance version, EOF datasets, ABSCO versions and the solar model.

## 250 3.5 Retrieval Algorithm

Inferring XCO$_2$ from measured radiances is an ill-posed inverse problem, which is referred to as performing a retrieval. The relationship between measurement and state is first modeled as

$$\boldsymbol{y} = F(\boldsymbol{x}) + \boldsymbol{\varepsilon}, \tag{11}$$

where data $\boldsymbol{y} \in \mathbb{R}^n$ is a radiance vector, unknown $\boldsymbol{x} \in \mathbb{R}^m$ is the state vector, $F : \mathbb{R}^m \to \mathbb{R}^n$ is the OCO-2 FP model and

$\boldsymbol{\varepsilon} \in \mathbb{R}^n$ is the measurement uncertainty. For completeness, we summarize the operational retrieval algorithm used in OCO-2





processing. The retrieval proceeds with solving the inverse problem by using Bayesian formulation, in which the additive error $\varepsilon$ and prior for $\boldsymbol{x}$ are assumed to be Gaussian, such that

$$\varepsilon \sim \mathcal{N}(0, \boldsymbol{S}_\varepsilon), \quad \boldsymbol{x} \sim \mathcal{N}(\boldsymbol{x}_a, \boldsymbol{S}_a). \tag{12}$$

The measurement error covariance matrix $\boldsymbol{S}_\varepsilon$ is assumed to be diagonal, with elements for each wavelength $j$ given by

$\quad \sigma_j^2 = k_1 y_j + k_2, \tag{13}$

where $k_1$ and $k_2$ are calibration parameters adjusted by the instrument calibration team. The a priori covariance is taken to be diagonal for non-$CO_2$ parameters, and the $CO_2$ profile is assumed to have a correlation structure shown in in Figure 2, which promotes continuous concentration profiles and limits the variability higher up in the atmosphere.

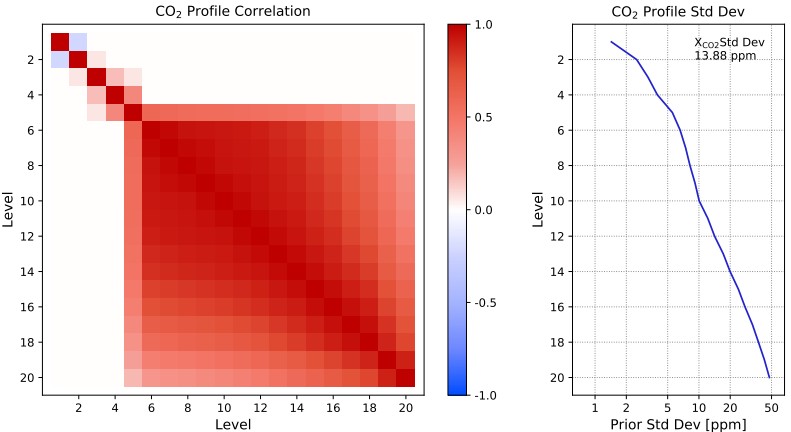

**Figure 2.** The a priori correlation matrix and standard deviation used for the $CO_2$ vertical profile in the OCO-2 retrieval. Vertical levels are ordered from the top of the atmosphere (Level 1) to the surface (Level 20).

The retrieval is operationally carried out using iterative gradient-based methods to solve for the maximum a posteriori
estimate, which is equivalent to minimizing the cost function

$$\widehat{\boldsymbol{x}} = \underset{\boldsymbol{x}}{\mathrm{argmin}} \left(\boldsymbol{y} - F(\boldsymbol{x})\right)^T \boldsymbol{S}_\varepsilon^{-1} \left(\boldsymbol{y} - F(\boldsymbol{x})\right) + \left(\boldsymbol{x} - \boldsymbol{x}_a\right) \boldsymbol{S}_a^{-1} \left(\boldsymbol{x} - \boldsymbol{x}_a\right) \tag{14}$$

This optimization problem is solved using the Levenberg-Marquardt algorithm, in which at iteration $i$ the state is updated according to

$$\left((1+\gamma)\boldsymbol{S}_a^{-1} + \boldsymbol{K}_i^T \boldsymbol{S}_\varepsilon^{-1} \boldsymbol{K}_i\right) dx_{i+1} = \left[\boldsymbol{K}_i^T \boldsymbol{S}_\varepsilon^{-1}(\boldsymbol{y} - F(\boldsymbol{x}_i)) + \boldsymbol{S}_a^{-1}(\boldsymbol{x}_a - \boldsymbol{x}_i)\right] \tag{15}$$

where $\gamma$ is a damping parameter and $\boldsymbol{K}_i$ is the Jacobian of $F(\boldsymbol{x})$ at iteration $i$. After each iteration, before updating the state, the effect of forward model non-linearity is assessed by computing the quantity





$$R = \frac{c_i - c_{i+1}}{c_i - c_{FC}}, \tag{16}$$

where $c_i$ is the value of the cost function (14) at iteration $i$, $c_{i+1}$ similarly at iteration $i+1$, and $c_{FC}$ is the cost function value assuming that $F(\boldsymbol{x}_i + d\boldsymbol{x}_{i+1}) = F(\boldsymbol{x}_i) + \boldsymbol{K}_i d\boldsymbol{x}_{i+1}$; that is, a linear update. Based on the value of $R$, one of the following is executed:

- $R \le 0.0001$: $\gamma$ is increased by a factor of 10. State is not updated.

- $0.0001 < R < 0.25$: $\gamma$ is increased by a factor of 10, $\boldsymbol{x}_{i+1} = \boldsymbol{x}_i + d\boldsymbol{x}_{i+1}$

- $0.25 < R < 0.75$: $\boldsymbol{x}_{i+1} = \boldsymbol{x}_i + d\boldsymbol{x}_{i+1}$

- $R > 0.75$: $\gamma$ is decreased by a factor of 2, $\boldsymbol{x}_{i+1} = \boldsymbol{x}_i + d\boldsymbol{x}_{i+1}$

After each non-divergent step, convergence is assessed by computing the error variance derivative (see Crisp et al. (2021) for details). The operational retrieval further provides an estimate for the posterior covariance as a Laplace approximation

$$\widehat{\boldsymbol{S}} = \left( \boldsymbol{K}^T \boldsymbol{S}_\varepsilon^{-1} \boldsymbol{K} + \boldsymbol{S}_a^{-1} \right)^{-1}, \tag{17}$$

together with the so-called *Averaging Kernel*

$$\boldsymbol{A} = (\boldsymbol{S}_a^{-1} + \boldsymbol{K}^T \boldsymbol{S}_\varepsilon^{-1} \boldsymbol{K})^{-1} \boldsymbol{K}^T \boldsymbol{S}_\varepsilon^{-1} \boldsymbol{K} \tag{18}$$

which can be interpreted as the sensitivity of the retrieved state $\widehat{\boldsymbol{x}}$ to the true atmospheric state $\boldsymbol{x}$. These quantities are important for downstream users of OCO data products, which highlights the value of producing closed-form Jacobians during data processing.

## 4 Forward Model Emulation

In this section, we will describe the practical implementation of our method laid out in Section 2 applied to the OCO-2 retrieval problem in Section 3. This includes data transformations and dimension reduction, training data generation, convergence of the optimizer in kernel parameter learning, and assessment of forward model output quality. We stress that in order to be implemented in an operational retrieval algorithm, the emulator is required to perform with superior accuracy. We ensure accurate performance by making sure that the error in predicted radiances, compared to FP outputs, is less than the radiance measurement error standard deviation. This way, any systematic errors in emulation will be masked by measurement noise, and retrieval performance using emulation will closely resemble that of using FP.





## 4.1 Data transformations

As GPs tend to perform worse with increasing input dimension, and because the standard GP formulation is developed for one dimensional outputs, we will need to reduce the dimension of both the atmospheric state and the radiance. For the atmospheric state $x$, we leverage the fact that OCO-2 measurements are made at 3 separate wave length bands, which leads to the state vector

having band-specific elements which can be ignored when dealing with other bands. This partition has been summarized in Table 1. Earlier work by Ma et al. (2019) considered cross-band correlations while emulating OCO-2 radiances, but the authors finally showed that the bands are distant enough from one another in wavelength space that they can be treated independently. With this insight, we proceed by constructing separate GPs for each band and using only the sensitive dimensions of $x$ as inputs. We further notice that the 20 element $CO_2$ profile is continuous and can be expressed as loadings obtained using

principal component analysis (PCA). The most straightforward way to do this is by truncated singular value decomposition (SVD) of the empirical covariance matrix of state vectors (Tukiainen et al., 2016). To accomplish this, we use a simulation distribution derived by Braverman et al. (2021) for one selected template as a basis for our experiments and perform SVD on the covariance matrix of this distribution. Analysis of singular value decay suggests that the $CO_2$ profile can be represented with just 4 principal components, which we collect to a matrix $P_x$ as the 4 leading singular vectors. We then project the $CO_2$ profile

to principal component space, and further standardize the states by using the mean and variance of the simulation distribution, leading to

$$\widetilde{x} = \frac{1}{\sigma_x} \left( \mathrm{diag}(P_x^T, \mathbb{I}_c)(x - [\mu_p, \mathbf{0}_c]) - \mu_x \right), \tag{19}$$

where $\mu_p$ is the $CO_2$ profile mean, $\mu_x$ and $\sigma_x$ are the state mean and state standard deviation, $\mathrm{diag}(\cdot)$ denotes a block diagonal matrix, $\mathbb{I}_c$ is a $c = m - 16$ dimensional unit matrix (as the profile is represented by 4 dimensions instead of original 20), and

$[\mu_p, \mathbf{0}_c]$ is a stacked vector of $CO_2$ profile mean and a $c$ dimensional zero vector.





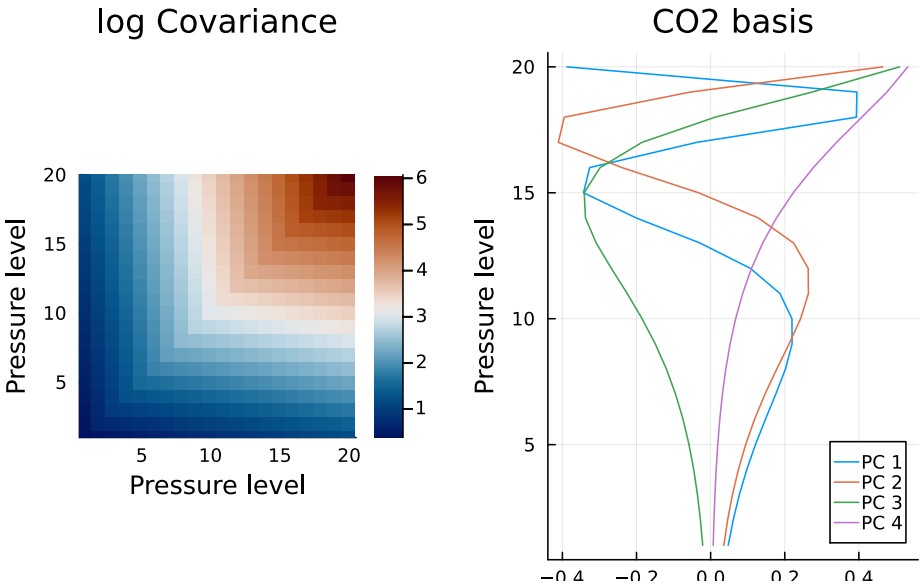

**Figure 3.** Left: the $CO_2$ profile covariance matrix of the simulation distribution used in this work. Right: 4 leading singular vectors from the SVD of the covariance matrix.

Next, we generate training data using a Sobol sequence (see Section 2.4). For this study, we can omit dispersion, EOF and SIF parts of the state vector (see Table 1) and fix them to the prior mean. This follows from the discussion in Section 3 focusing on monochromatic radiances. Omitting dispersion simplifies of computations as the wavelength grid would otherwise shift, making SVD for radiance dimension reduction hard. Ma et al. (2019) solved this problem by employing Functional Principal

Component Analysis, while we can proceed with ordinary SVD. The Empirical Orthogonal Functions (EOFs) are included in the operational retrieval to reduce fit residuals and therefore make convergence analysis easier. These has no direct impact on our study and can be safely omitted. Furthermore, the SIF parameters are fit on the O2 band only as part of the instrument effects, and we do not include them in the emulation for this reason. As is evident from equation (9), the measurement geometry has a significant impact on the output of the FP model. For this reason we include three extra parameters, $\theta, \theta_0$, and $\varphi - \varphi_0$

to our training data vector. Sufficient and realistic limits to these parameters are obtained from the simulation distribution of Braverman et al. (2021) by considering a $4\sigma$ interval around the mean values. In all, we now have $m = 4 + 21 + 3 = 28$ for input space, coming from profile PCs, other included state vector elements and geometry. We create a Sobol sequence of 20 000 points for training, and scale all dimensions of the hypercube to $[-4, 4]$, corresponding to 4 standard deviations in the normalized $\widetilde{x}$ basis. We further obtain the training data set in original space by reversing the transformation (19).

Training data $Y$ (radiances) are obtained by evaluating the FP model on each $x$ from the scaled Sobol sequence. For this work, we choose a single realistic land nadir measurement to represent physical parameters not included in state vector $x$. We perturb sampling geometry to reflect relevant solar and instrument angles. For a real-world application this approach can be extended to include different scenes and other location-dependent parameters. To obtain the labels $z$, we similarly perform




truncated SVD on the radiances $\boldsymbol{Y}$ separately on each wavelength band $B \in [O2, WCO2, SCO2]$, and collect the leading $n_B$
singular vectors in matrices $\boldsymbol{P}_B$. The four leading singular vectors for each band are presented in Figure 4. With additional
standardization of the variables, we obtain the following transformations for each wavelength band $B$:

$$\widetilde{\boldsymbol{z}} = \frac{1}{\boldsymbol{\sigma}_z}(\boldsymbol{P}_B^T (\boldsymbol{y}_B - \boldsymbol{\mu}_B) - \boldsymbol{\mu}_z) \tag{20}$$

where $\boldsymbol{\mu}_B$ is the radiance mean, and $\boldsymbol{\mu}_z, \boldsymbol{\sigma}_z$ are the principal component mean and standard deviation for band $B$.

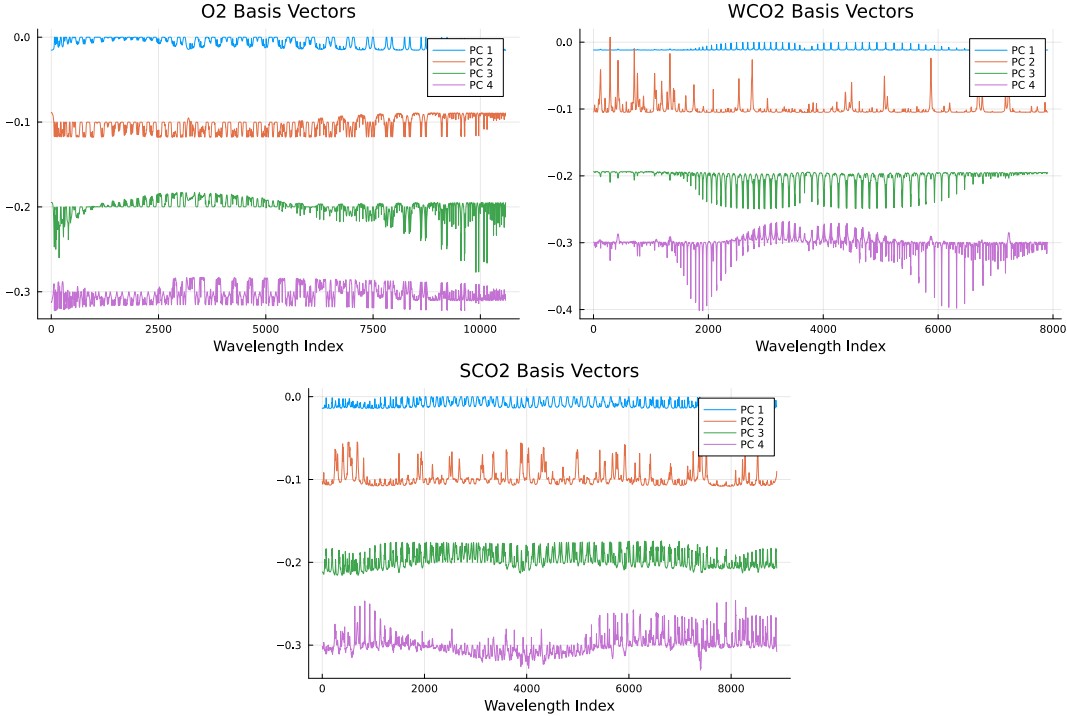

**Figure 4.** Leading 4 basis vectors obtained from the SVD (principal components, PC) for radiances $\boldsymbol{y}$ for each band: O2, Weak CO2 (WCO2)
and Strong CO2 (SCO2). Basis vectors 2-4 are offset for illustration purposes.

The quality of this approximation is assessed by plotting the reconstruction $\boldsymbol{P}_B \boldsymbol{P}_B^T (\boldsymbol{y}_B - \boldsymbol{\mu}_B) + \boldsymbol{\mu}_B$ over a heldout dataset
not used in computing the SVD. We illustrate in the upper panel of figure 5 the distrubution of relative reconstruction error from
this dataset. We have further applied the instrument function to each residual and further divided them by the measurement
error standard deviation given by equation (13). This metric is justified by the rationale that if the reconstruction error is less
or comparable to measurement error on the radiances, no significant amount of information is lost.





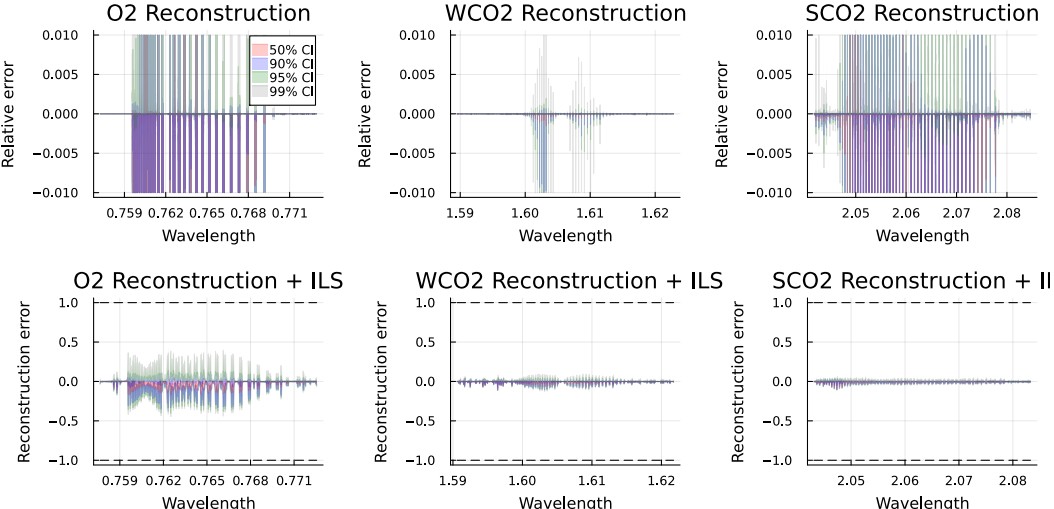

**Figure 5.** Upper row: distribution of relative reconstruction error for monochromatic radiances on the $O_2$, $WCO_2$ and $SCO_2$ bands. Lower row: distribution of reconstruction error for all bands after applying the instrument function and dividing by measurement error standard deviation. Shading represents 50% (red), 90% (blue), 95% (green) and 99% (gray) confidence intervals.

The final emulator $g(\boldsymbol{x})$ can now be summarized in Figure 6.


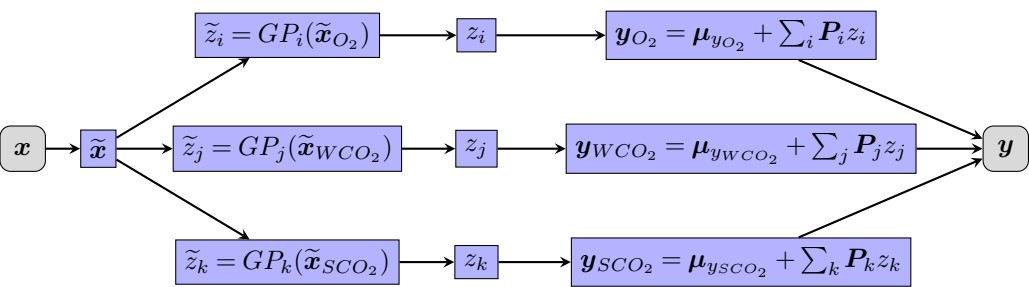

**Figure 6.** Diagram showing the step-by-step process of emulator evaluation.

where $GP_i(\widetilde{x}_B)$ is the GP prediction given by equation (1) and the indices $i, j, k$ run through the number of principal components included in a given band. The effect of this choice will be examined further later in this work. When evaluating the emulator, each index is independent and can then be computed in parallel.

## 4.2 Training

Having obtained training data $\widetilde{X}, z$, we can now proceed to optimize the kernel parameters as described in section 2.3. We prescribe an individual GP per output parameter $z_i$. We have $N = 20000$ for training data size, and we set $M = 100$ for mini-batch size, $p = 5$ for the number of prediction points per mini-batch, and run the ADAM optimizer for 5000 iterations with a



small learning rate. We initialize all other parameters at 1, except for linear component weight at 0 and the nugget at 1e-6. For $\widetilde{x}$. As outlined in Section 3.3, we further reduce the dimension of the input space by selecting only the indices that a given

wavelength band is sensitive to, given by Table 1.

For testing the performance of the algorithm, we draw a random sample $X^{\text{test}}$ from the simulation distribution as independent test data, which is then used to evaluate the FP model to create radiances $Y^{\text{test}}$. For test data, we fix dispersion, EOFs and SIF at prior values as before. Example behavior of the loss function together with evolution of the kernel parameter values and true vs predicted $z$ values for is shown in Figure 7. The distribution of true vs predicted $z$ values for each component on each

wavelength band is illustrated in Figure 8.

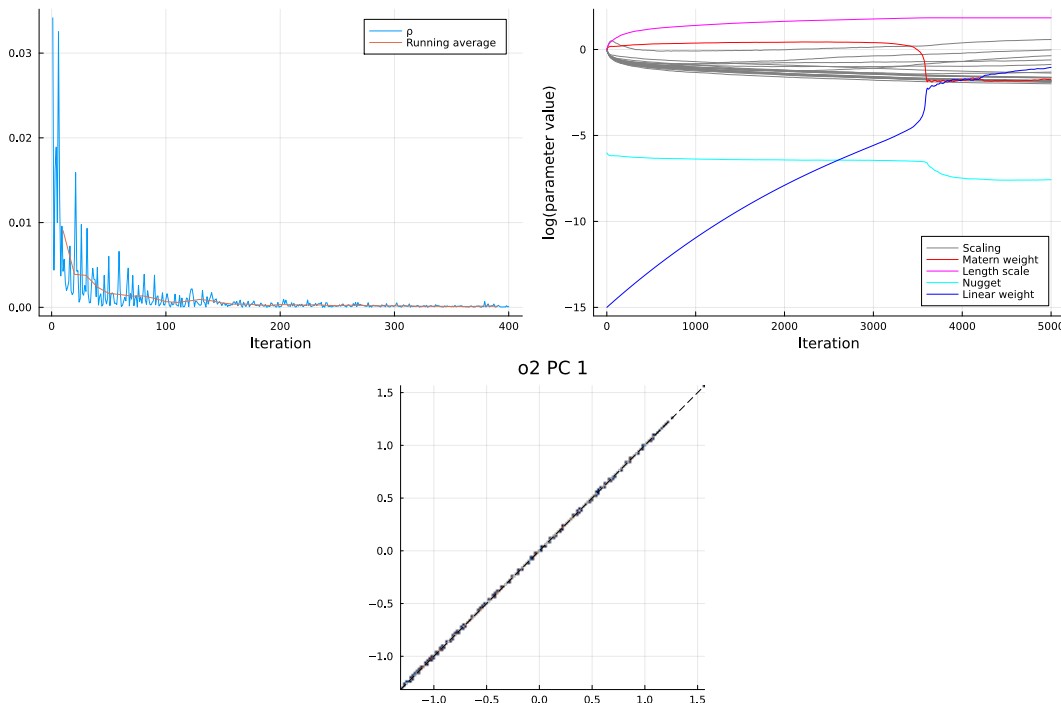

**Figure 7.** Example training performance for the first principal component of the $O_2$ band. Upper left: loss function values for the first 400 iterations of parameter learning. Blue line depicts the cost function value per iteration. Red line depicts a cumulative running average of the cost function. Upper right: evolution of the kernel parameters as function of iteration. Lower center: true vs predicted $z$ values over a withheld test set for the first component of $O_2$ band.





**Figure 8.** True versus predicted values for 10 radiance principal components on each wavelength band.

## 4.3 Predictive performance

Finally, we assemble the predicted $z$ values back to radiances and compute the relative differences with the test data, shown in the upper panel of figure 9. On the lower panels, as before, we apply the instrument function to these residuals and divide by





measurement error standard deviation to underline that desired performance would be to make less prediction error than is the
measurement error.

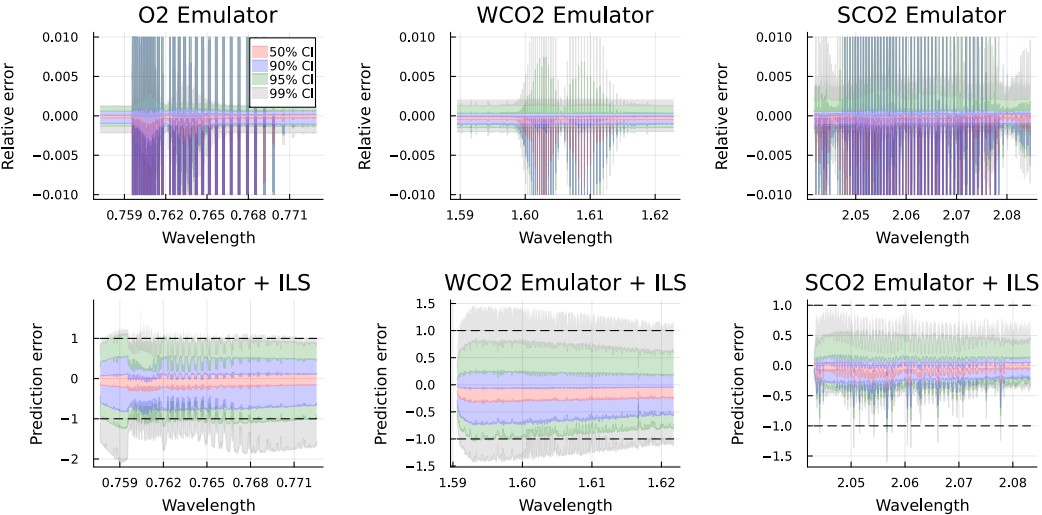

**Figure 9.** Upper row: distribution of relative prediction error for monochromatic radiances on the $O_2$, $WCO_2$ and $SCO_2$ bands. Lower row: distribution of prediction error for all bands after applying the instrument function and dividing by measurement error standard deviation. Shading represents 50% (red), 90% (blue), 95% (green) and 99% (gray) confidence intervals.

After constructing the emulator obtaining radiances as outputs, we can further apply equation (4) to compute the Jacobians $\frac{d}{d\widetilde{x}}z$. We can then reverse the normalizing transformations on both $\widetilde{x}$ and $y$ and further apply the instrument functions to our Jacobians to get back to the operational observation units. The Jacobians obtained by evaluating both FP and emulator on an example state vector together with the resulting profile averaging kernels are shown in Figure 10. We note that we have

normalized the Jacobians and averaging kernels by maximum values of each row in the matrix for visual clarity. Although not perfectly similar, we conclude that these two outputs share significant similarity. The main difference on the averaging kernels mainly results from the choices on modeling concentration profiles by principal component loadings.





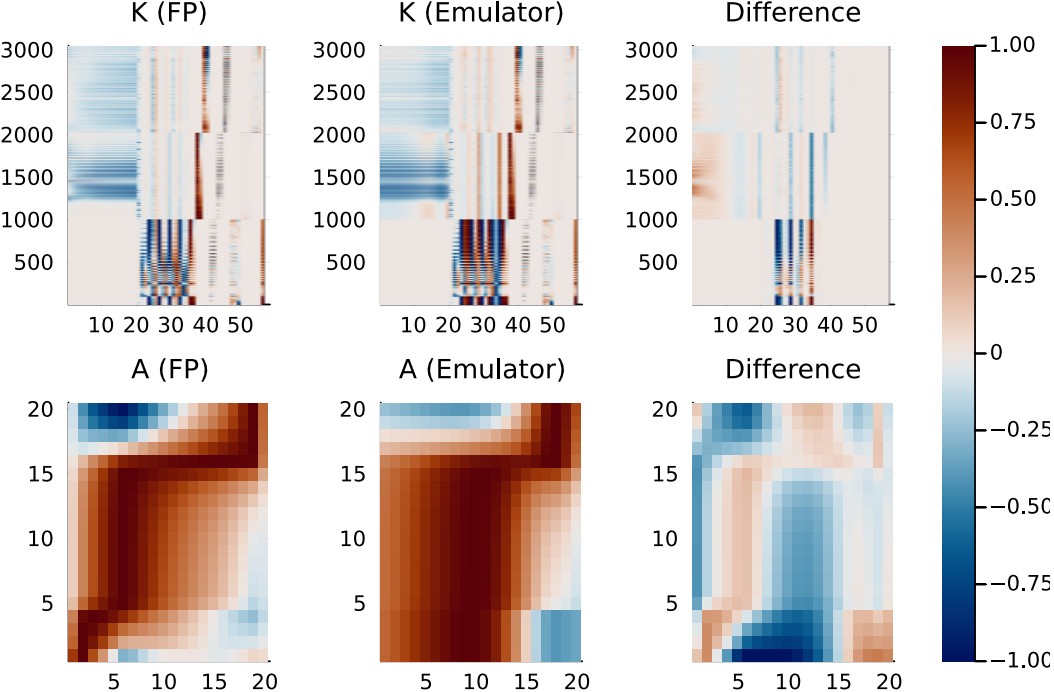

**Figure 10.** Normalized Jacobians $K$ and profile averaging kernels $A$ from both the FP and emulator, together with the corresponding differences.

As noted in previous work by Ma et al. (2019), an emulator provides substantial appeal in terms of computational efficiency. For the current work, the average computational times for model evaluation and Jacobians are summarized in Table 2 on a 2023 MacBook Pro. Three cases are contrasted: the standard ReFRACtor FP evaluation, the emulator for monochromatic radiances plus ILS, and the emulator alone.

**Table 2.** Evaluation times of radiative transfer (RT) model and related Jacobian, comparing the ReFRACtor implementation, monochromatic emulator with instrument line shape (ILS) and other spectral corrections, and monochromatic emulator only.

|  | RT [s] | RT + Jacobian [s] |
|---|---|---|
| ReFRACtor | 33.45 | 55.26 |
| Emulator + ILS | 2.06 | 2.17 |
| Emulator | 0.05 | 0.19 |

## 4.4 Faster Research Version

In recent years, the uncertainty quantification and statistics community has benefited enormously by utilizing the surrogate model by (Hobbs et al., 2017) to explore the OCO-2 retrieval in numerous applications (Brynjarsdóttir et al. (2018), (Lamminpää et al., 2019), Nguyen and Hobbs (2020), Hobbs et al. (2021), Patil et al. (2022)). We remark that for similar purposes,





our emulator can be used as an even faster surrogate. As we see from Table 2, the majority of the computational cost for the emulator comes from the instrument effects which is part of the ReFRACtor software. If one is not interested in including the effects of dispersion, SIF, and EOF's during the retrieval, we notice from equation (10) that instrument corrections to RT amount to multiplication, addition, and convolution, which is associative wrt. multiplication. We can then write the emulator

as

$$g(\boldsymbol{x}) = ILS\left(\widehat{\boldsymbol{P}}\eta(\boldsymbol{x})\right) = ILS\left(\widehat{\boldsymbol{P}}\right)\eta(\boldsymbol{x}), \tag{21}$$

where $g(\boldsymbol{x})$ is the overall emulator, $ILS()$ is a function applying the instrument corrections from equation (10), $\widehat{\boldsymbol{P}}$ is a projection matrix consisting of radiance basis functions that corresponds to transforming predicted labels $\boldsymbol{z}$ back to radiances $\boldsymbol{y}$ following the last step in figure 6, and $\eta(\boldsymbol{x})$ is the emulator predicting labels $\boldsymbol{z}$ from inputs $\boldsymbol{x}$. Done this way, we can evaluate

the instrument corrections on the basis vectors once, after which OE or MCMC can proceed an order of magnitude faster (according to table 2).

## 5 Retrievals using the Emulator

We are now ready to compare the performance of the emulator against the FP model when performing simulated retrievals. After obtaining the minimizer $\widehat{\boldsymbol{x}}$ and a Laplace approximation of posterior covariance, $\widehat{\boldsymbol{S}}$, the quantity of interest is further

given by multiplying the $CO_2$ profile by the pressure weighting function $\boldsymbol{h}$ that puts an appropriate weight for each pressure level, resulting in

$$\text{XCO}_2 = \boldsymbol{h}^T\widehat{\boldsymbol{x}}_{1:20}. \tag{22}$$

The reported uncertainty coming with the QoI is given by

$$\text{XCO2}_{\text{uncert}} = \sqrt{\boldsymbol{h}^T\widehat{\boldsymbol{S}}_{1:20,1:20}\boldsymbol{h}}. \tag{23}$$

We present two test cases for assessing retrieval performance of our emulator. First, we create synthetic observations by evaluating the FP model on our test set of states $\boldsymbol{x}$ and adding a realization from the Gaussian noise distribution:

$$\boldsymbol{y}_{\text{test}} = F(\boldsymbol{x}, \boldsymbol{b}) + \boldsymbol{\varepsilon}, \tag{24}$$

where $\boldsymbol{\varepsilon} \sim N(0, \boldsymbol{S}_{\boldsymbol{\varepsilon}})$. Second, we follow the methods outlined in Braverman et al. (2021) to further corrupt the simulated measurement by realistic *Model Discrepancy* (MD) adjustment, given by

$$\boldsymbol{y}_{\text{test}} = F(\boldsymbol{x}, \boldsymbol{b}) + \boldsymbol{\varepsilon} + \boldsymbol{\delta}, \tag{25}$$



where $\boldsymbol{\delta} \sim N(\boldsymbol{\mu}_\delta, \boldsymbol{S}_\delta)$. The shape of this adjustment is illustrated in Figure 11. As noted by the authors, model discrepancy as presented here is a statistical representation of forward modeling mismatches so that our simulated measurements would better correspond to real data.

We then perform XCO$_2$ retrievals both using the Full Physics model $F(\boldsymbol{x})$ and the emulator $g(\boldsymbol{x})$ following the algorithm laid
out in Section 3. Results for retrieved XCO$_2$ for both cases with and without MD are illustrated in Figure 12. The corresponding XCO$_2$ uncertainty values are compared in Figure 13. We conclude that using the emulator in place of FP model in retrieval preserves the accuracy and replicates same biases as FP, while having good correlation with each other. On the other hand, the output uncertainty estimates seem to not correspond to each other, and further analysis on this output will be required in future research work.

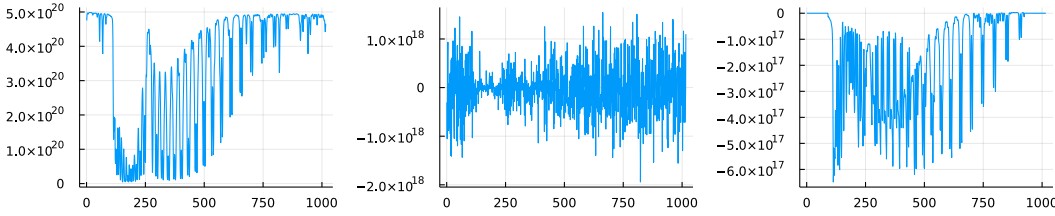

**Figure 11.** Left: example $O_2$ band radiance. Middle: realization from the noise distribution $N(0, \boldsymbol{S}_\varepsilon)$. Right: realization from the model discrepancy distribution $N(\mu_\delta, \boldsymbol{S}_\delta)$. Units for all panels are W m$^{-2}$ sr$^{-1}$ $\mu$m$^{-1}$, the units of radiance for OCO-2.





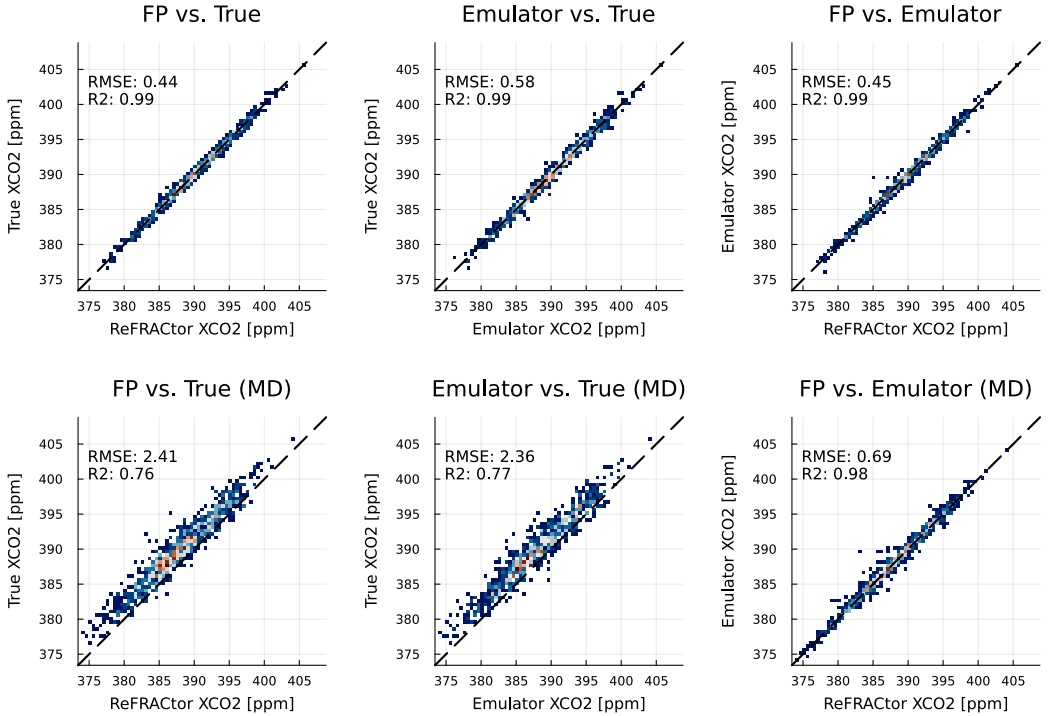

**Figure 12.** Retrieved $XCO_2$ over heldout dataset using FP and emulator. Upper row: Full Physics (left), Emulator (middle) and comparison of the two (right). Lower row: similarily, but with added model discrepancy in observed data. RMSE (Root mean square error) describes bias of the retrievals while $R^2$ value is included to assess correlation between quantities of interest.

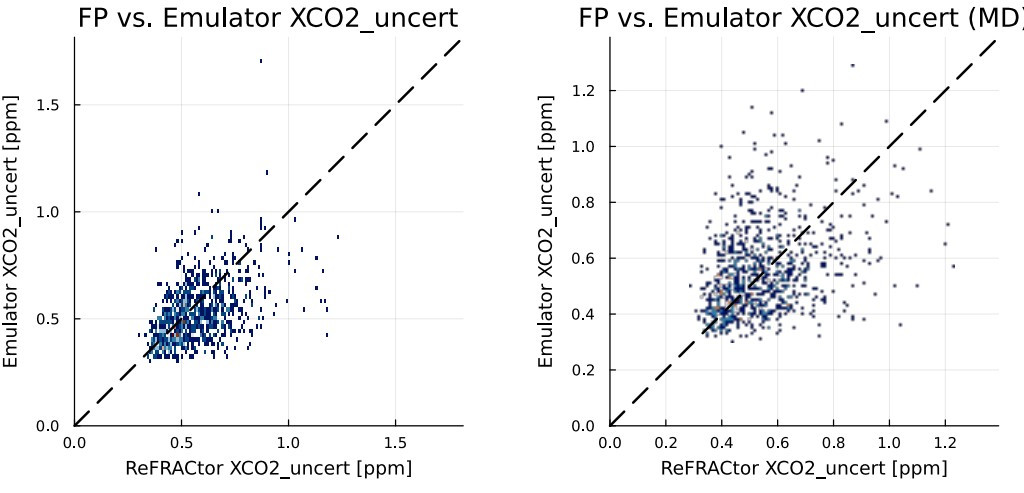

**Figure 13.** Scatter plots of retrieval $XCO_2$ uncertainty over heldout dataset using FP and emulator. Left: no MD in observations. Right: MD included in observations.

### 5.1 Effect of PCA dimensionality

Previously in this work we have not prescribed a certain number of principal components to use in radiance dimension reduction. Figure 14 illustrates the retrieved $XCO_2$ root mean square error (RMSE) and mean absolute error (MAE) against the true known value, together with 15 illustrating radiance reconstruction and prediction RMSE and MAE similarly to Figures 5 and 9, all as a function of number of PCs used. We can collectively deduce that using more than 25 principal components per band does not yield any additional performance benefits. We remark that compared to the earlier work by Ma et al. (2019), who argued for 1-3 principal components per band, our results show that many more components are needed for accurate retrievals. This highlights the importance of empirically checking the effect of dimensionality reduction and not relying on rules-of-thumb such as conserving 95% of variability.

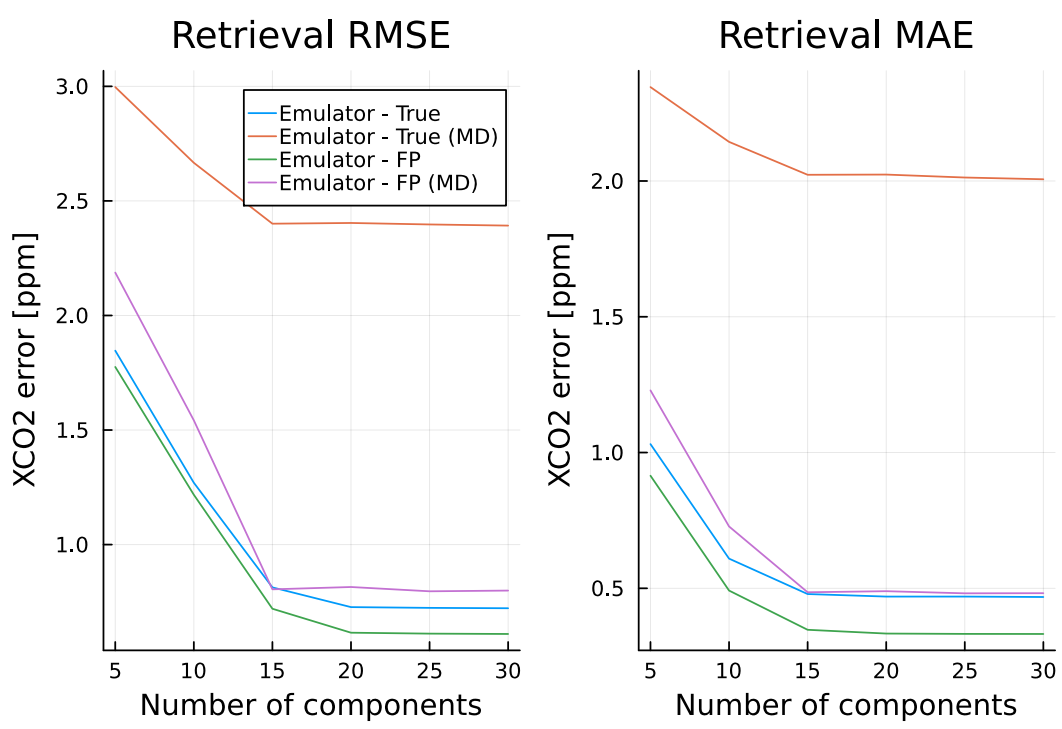

**Figure 14.** RMSE and MAE as a function of number of principal components used per band in radiance dimension reduction for $XCO_2$ retrievals, both with and without MD, over a heldout test dataset.

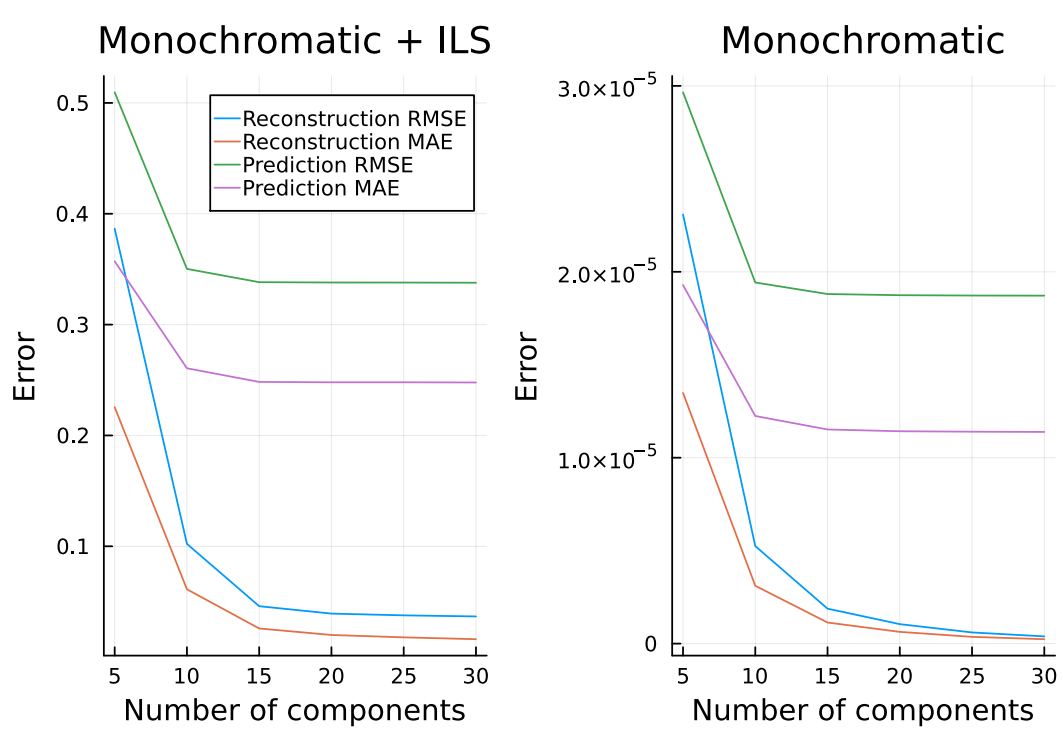

**Figure 15.** RMSE and MAE as a function of number of principal components used per band in radiance dimension reduction reconstructions and predictions, all over a heldout test dataset.





## 5.2 Effect of Aerosol types

To asses the effect of changing the dominant aerosol types on the performance of the retrievals, we repeat the training and retrieval procedure described in this section with two separate pairs of dominant aerosol types. First, we consider dust (DU) and sea salt (SS), and secondly, DU and sulphate (SO). These are among the most common aerosol combinations encountered in the OCO-2 operations. We repeat the retrievals for both cases with additional MD adjustment as before. Results for this experiment are summarized in figure 16. We conclude that the proposed method is robust to changing physical conditions, which indicates fitness for further operational integration.

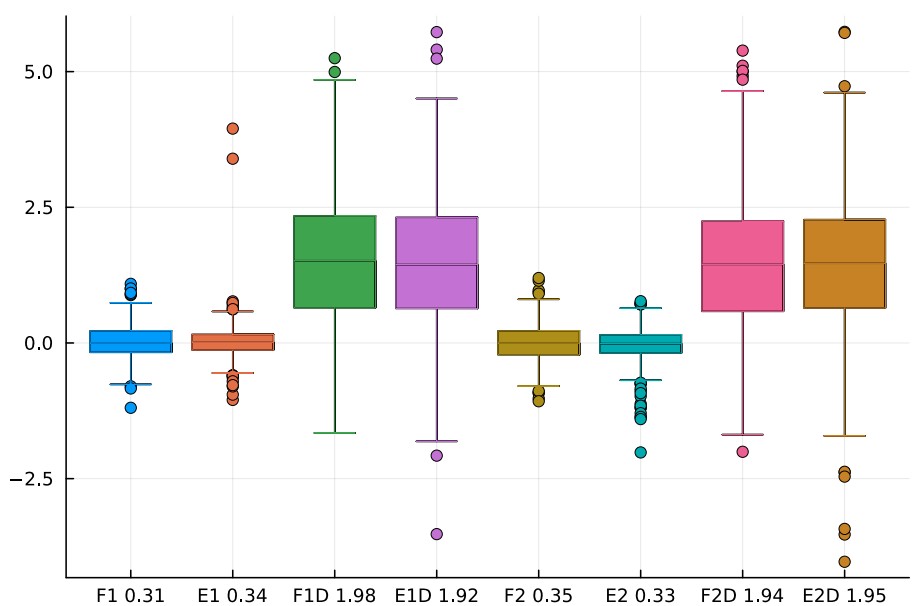

**Figure 16.** Difference in ppm between true and retrieved $XCO_2$ from simulated measurements with different dominant aerosol species, in ppm. Symbols on x-axis denote the specifics of given experiment: F = Full Physics, E = Emulator. 1 = DU and SS aerosols, 2 = DU and SO aerosols. D = With model discrepancy, followed by a number denoting mean error in ppm.

## 6 Conclusions

In this work, we have constructed and implemented a fast and accurate forward model emulator for the ReFRACtor implementation of the OCO-2 full physics forward model. The emulator produces closed form Jacobians, and as such gives a convenient way of performing $XCO_2$ retrievals. We have demonstrated the accuracy of these retrievals, and analyzed the effect of PCA

dimension, aerosol types and model discrepancy on the retrieval. All these tests indicate robustness and excellent reliability of our method, and offer an encouraging proof of concept for future operational implementation with latest ACOS algorithm and real world OCO-2 data.



This work has significantly advanced the Kernel Flows methodology (Owhadi and Yoo, 2018) by including a cross-validation based training strategy using RMSE cost function and new strategy for mini-batching. With this method, we have achieved a relative error of less than 1% which on its own is a significant improvement from the point of view of operator learning. Our approach is computationally fast and, when training data set is properly engineered, performs consistently withing the span of training data. Compared with our ability to compute Jacobians in closed form, our approach holds a potential to solve current and future data processing issues in atmospheric remote sensing stemming from computationally intensive forward models.

While Gaussian Process methods offer an attractive means to include uncertainty propagation in emulation pipeline, our tests have shown that the predicted posterior standard deviation given by GPs was not adequate in providing reliable coverage of true labels after prediction. This is likely due to Kernel Flows method's focus on optimizing the posterior mean prediction without assessing the prediction uncertainty. This could easily be remedied by including a uncertainty tuning penalty in the KF loss function. Another disclaimer comes from evaluations of retrieval uncertainty on XCO2: our method was not agreeing with operational OE. This does not mean our estimates were better or worse, and further research is needed in calibrating retrieval uncertainties.

A logical next step would be to implement the GP emulator for operational ACOS forward model instead of ReFRACtor, which requires closer collaboration with the OCO algorithm team. After demonstration on OCO-2, our approach is directly applicable for a myriad of other satellite missions. We note that future work will have to deal with training data design that was simplified in this work. Assessing different temporal and spatial variability in forward model parameters together with feasible distributions of state vectors will be key in this design effort. These efforts might benefit from including a cost-benefit analysis on training a global model usable everywhere versus, for example, re-training the emulator for sufficiently specified spatio-temporal datasets.

*Code and data availability.* Code and data are available at an OSF repository (Lamminpää, 2024). The software depends on ReFRACtor and ReFRACtorUQ git repositories which are freely available as well.

*Author contributions.* O.L.: Project administration, Conceptualization, Methodology, Software, Formal analysis, Data curation, Writing – Original Draft, Writing - review & editing, Visualization. J.S.: Conceptualization, Methodology, Software, Writing - original draft, Writing - review & editing. J.H.: Project administration, Conceptualization, Methodology, Supervision, Software, Data curation, Writing - review & editing. J.M.: Methodology, Software, Data curation, Writing - original draft. A.B.: Conceptualization, Methodology, Writing - review & editing. H.O.: Conceptualization, Methodology, Writing - review & editing.

*Competing interests.* The authors declare no competing interests.





*Acknowledgements.* The research described in this paper was performed at the Jet Propulsion Laboratory, California Institute of Technology, under contract with NASA. The authors thank Pulong Ma and Chris O'Dell for helpful guidance.

## Appendix A: Closed Form Jacobians

To obtain a closed for equation for the Jacobians used in the $XCO_2$ retrievals, we must explicitly compute the term $\frac{d}{d\boldsymbol{x}^*}\boldsymbol{\Gamma}(\boldsymbol{x}^*, \boldsymbol{X})$

in equation (4). To accomplish this, we compute the partial derivative of the kernel function (3) wrt. the first input:

$$\frac{\partial}{\partial \boldsymbol{x}} k(\boldsymbol{x}, \boldsymbol{x}') = \frac{\partial}{\partial \boldsymbol{x}} \left[ \alpha_1 \left( 1 + \frac{\sqrt{3}}{l} \|(\boldsymbol{x} - \boldsymbol{x}')\|_{\mathcal{W}} \right) \exp\left( -\frac{\sqrt{3}}{l} \|(\boldsymbol{x} - \boldsymbol{x}')\|_{\mathcal{W}} \right) + \alpha_2 (\mathcal{W}\boldsymbol{x})^T (\mathcal{W}\boldsymbol{x}') \right] \tag{A1}$$

$$= \frac{\partial}{\partial \boldsymbol{x}} \left[ \alpha_1 \exp\left( -\frac{\sqrt{3}}{l} \|(\boldsymbol{x} - \boldsymbol{x}')\|_{\mathcal{W}} \right) \right] + \frac{\partial}{\partial \boldsymbol{x}} \left[ \alpha_1 \left( \frac{\sqrt{3}}{l} \|(\boldsymbol{x} - \boldsymbol{x}')\|_{\mathcal{W}} \right) \exp\left( -\frac{\sqrt{3}}{l} \|(\boldsymbol{x} - \boldsymbol{x}')\|_{\mathcal{W}} \right) \right] \tag{A2}$$

$$+ \frac{\partial}{\partial \boldsymbol{x}} \left[ \alpha_2 (\mathcal{W}\boldsymbol{x})^T (\mathcal{W}\boldsymbol{x}') \right] \tag{A3}$$

$$= \frac{\partial \boldsymbol{d}}{\partial \boldsymbol{x}} \frac{\partial}{\partial \boldsymbol{d}} \left[ \alpha_1 \exp\left( -\frac{\sqrt{3}}{l} \boldsymbol{d} \right) \right] + \frac{\partial \boldsymbol{d}}{\partial \boldsymbol{x}} \frac{\partial}{\partial \boldsymbol{d}} \left[ \alpha_1 \left( \frac{\sqrt{3}}{l} \boldsymbol{d} \right) \exp\left( -\frac{\sqrt{3}}{l} \boldsymbol{d} \right) \right] + \frac{\partial}{\partial \boldsymbol{x}} \left[ \alpha_2 (\mathcal{W}\boldsymbol{x})^T (\mathcal{W}\boldsymbol{x}') \right] \tag{A4}$$


$$= \frac{\partial \boldsymbol{d}}{\partial \boldsymbol{x}} \left[ -\alpha_1 \left( \frac{\sqrt{3}}{l} \right) \exp\left( -\frac{\sqrt{3}}{l} \boldsymbol{d} \right) \right] + \frac{\partial \boldsymbol{d}}{\partial \boldsymbol{x}} \left[ \alpha_1 \left( \frac{\sqrt{3}}{l} \right) \exp\left( -\frac{\sqrt{3}}{l} \boldsymbol{d} \right) - \alpha_1 \left( \frac{\sqrt{3}}{l} \right)^2 \boldsymbol{d} \exp\left( -\frac{\sqrt{3}}{l} \boldsymbol{d} \right) \right] \tag{A5}$$

$$+ \left[ \alpha_2 (\mathcal{W})^T (\mathcal{W}\boldsymbol{x}') \right] \tag{A6}$$

$$= \frac{\partial \boldsymbol{d}}{\partial \boldsymbol{x}} \left[ -\alpha_1 \left( \frac{\sqrt{3}}{l} \right)^2 \boldsymbol{d} \exp\left( -\frac{\sqrt{3}}{l} \boldsymbol{d} \right) \right] + \left[ \alpha_2 (\mathcal{W})^T (\mathcal{W}\boldsymbol{x}') \right] \tag{A7}$$

$$= \frac{\mathcal{W}(\mathcal{W}(\boldsymbol{x} - \boldsymbol{x}'))}{\boldsymbol{d}} \left[ -\alpha_1 \left( \frac{\sqrt{3}}{l} \right)^2 \boldsymbol{d} \exp\left( -\frac{\sqrt{3}}{l} \boldsymbol{d} \right) \right] + \left[ \alpha_2 (\mathcal{W})^T (\mathcal{W}\boldsymbol{x}') \right] \tag{A8}$$

$$= \mathcal{W}^2 \left[ -\alpha_1 \left( \frac{\sqrt{3}}{l} \right)^2 \exp\left( -\frac{\sqrt{3}}{l} \|(\boldsymbol{x} - \boldsymbol{x}')\|_{\mathcal{W}} \right) (\boldsymbol{x} - \boldsymbol{x}') + \alpha_2 \boldsymbol{x}' \right] \tag{A9}$$

After computing $\frac{\partial}{\partial \boldsymbol{x}} k(\boldsymbol{x}, \boldsymbol{x}')$, we get $\frac{d}{d\boldsymbol{x}^*} \boldsymbol{\Gamma}(\boldsymbol{x}^*, \boldsymbol{X})$ element-by-element with $\boldsymbol{x}$ being the new input and $\boldsymbol{x}'$ a training data point. The final Jacobian is then obtained by computing $\frac{d}{d\boldsymbol{x}^*} z^*$ via (4) and reversing transformations (19) and (20).





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
