# Peer review of "Forward Model Emulator for Atmospheric Radiative Transfer Using Gaussian Processes And Cross Validation"

_Atmospheric Measurement Techniques, 2024_

## Author Response (AR1)

**Reviewer Comments And Replies**

**Reviewer 1**

SUMMARY

The authors of this study propose a data-driven surrogate of the forward model needed to relate atmospheric state to CO2 measurements in the context of the OCO-2 satellite mission. The approach is based on a powerful regression-based technique known as Gaussian Processes (GPs), which can approximate continuous functions to arbitrary accuracy (i.e., they enjoy the universal function approximation property). The paper describes and implements an enhanced GP emulator inspired by some recent developments in the field (namely, kernel flows with cross validation), and further incorporates dimension reduction techniques to efficiently deal with large volumes of data. The resulting algorithm is tested on synthetic datasets and shown to effectively recover the true forward model within a 1% relative error. Findings are encouraging and pave the way for future implementations on real CO2 datasets.

This is a very well written paper that documents all necessary technical aspects of the new GP surrogate model and offers a detailed evaluation of its performance. The authors provide an excellent motivation of the research topic, raising awareness of both the climate impacts of CO2 levels and the difficulties in measuring the concentrations of this trace gas. A particularly strong feature of the paper are the carefully justified sensitivities to various design choices in the GP algorithm; it was insightful to see how rule-of-thumb criteria for selecting an appropriate number of PC components may not be adequate (Section 5.1) and how GP predictions are robust to various dominant aerosol species (Section 5.2). Along similar lines, it was important to emphasize how domain knowledge can play a crucial role for optimizing the accuracy and cost of GP predictions (e.g., the justifications for wavelength band separation).

On account of this positive feedback, I recommend this paper to be accepted for publication in AMT after the authors address my comments below. Most of them are fairly minor and line-specific, but the next section discusses a few more general suggestions pertaining to the manuscript's presentation that will help further increase its impact.

GENERAL COMMENTS

1. **Methodology** (in particular L66-L84, but also other related parts of the paper discussing the novelty of your work): Given the choice of a journal, I expect the primary audience of this article to be experts in measurement techniques who might not be fully aware of all recent advances in the statistical and ML literature. Hence, I advise the authors to make their presentation slightly less technical, and instead focus on some of the practical advantages of using their enhanced GP emulator for retrieval problems. In simple terms, why is the GP method better than standard neural networks? What aspect of the retrieval problem can kernel flows help with — accuracy, computational speed, both? In what sense is the the new cross-validation based training considered an extension of kernel flow techniques (i.e., what is the intuition behind a "flow" in cross validation)? How do Gaussian processes handle bounded datasets ($CO_2$ concentrations are non-negative)? What is the alternative to Gaussian processes and neural networks for emulating complex forward models like ReFRACtor? These are some of the questions practitioners would like to know before adopting your GP-based methodology.

2. **Technical comments on the GP presentation** (particularly in terms of Eq. 1 and related text): The GP regression in Eq. (1) has a very similar structure to that of Optimal Interpolation (OI; e.g., page 157 of Kalnay 2003's textbook "Atmospheric modeling, data assimilation and predictability") and related iterative Optimal Estimation (OE) algorithms (e.g., Eq. 15 later in the manuscript). Specifically, the terms pre-multiplying "z" look similar to the OI weight matrix $W = BH^T (HBH^T + R)^{-1}$, where B is the "background" covariance associated with z, R (which is $\sigma \Gamma$ in your paper) is the measurement error covariance matrix and H (K in your notation) is the Jacobian of the observation operator (forward model in your nomenclature). To improve the reader's intuition about GP models, it will be nice to include a short discussion on these connections as well as the underlying differences with GP (e.g., OI/OE do not have kernel evaluations in the covariance matrix definitions).

3. **Figure captions and in-text figure interpretations**: I sometimes found that figure captions do not provide a sufficient description of their content (e.g., Fig. 6). Related to that issue, in-text interpretation of some figures was fairly minimal relative to their complexity (see next section for specific examples). I think the authors should add a few more details to make this paper more self-contained.

SPECIFIC COMMENTS

**L28**: It will be helpful to include references showing the lack of $CO_2$ measurements in the Tropics.

**L31-L32**: Please elaborate on what total column mole-fraction $CO_2$ is.

**L46**: If you are aware how many soundings are discarded in the process, include this in the text.

**L68**: What do you mean by model atmospheres?

**L90**: Please ensure consistent notation: radiances are denoted by y here, but Section 2.1 uses z for the labels.

**Eq. (2)**: I assume there exists a more general expressions for the posterior covariance?

**L117**: Either provide references for these kernels or write them explicitly.

**L119-L120**: Explain why differentiating the kernel function is important; this comes slightly later on L125, but I think it will better to expand on this point earlier.

**L125**: I would say "Taking the derivative of eq. (1) wrt to x*" here.

**Eq. (5)**: Either provide a reference for this likelihood function or explain/derive how it can be obtained.

**L145**: $L^2$ space not explicitly defined, it might be better to continue using the notion of mean squared errors to improve readability.

**Eq. (6)**: It is not clear whether this expression can be derived from Eq. (1). The text around L150 suggests a rank-based approximation is used, but the authors should clarify the connection to the original GP regression in Eq. (1).

**L156**: Clarify what you mean by a k-norm.

**L160-L166**: It will be good to discuss how the ability to compute the kernel derivatives in closed form factors into the automatic differentiation procedure.

**Line 1 of Algorithm 1**: Shouldn't R be initialized according to the \theta_1 values, which are set to 1? That is, the initial loss should be consistent with the initial choice of GP parameters.

**Section 2.4**: What other options are there for generating the parameters? What motivated your choice for using Sobol sequences?

**FP model (L190-L215)**: It was not very clear to me how Eqs. (9) and (10) participate in the general forward model described by Eq. (8). For example, is F (loosely) defined as the sum of Eqs. (9) and (10)?

**Fig. 1**: In the figure caption, please clarify where these aerosol types are used.

**L285-L287**: At this point a significant time period has passed since you last described what K is; to fully appreciate the usefulness of closed-form Jacobians of the forward model (i.e., the matrix K), it will be helpful to remind them again what K is.

**L204**: Define loadings.

**L307**: What do you mean by template?

**Eq. (19)**: diag(*,*) operation not defined, only diag(*) discussed. I assume diag(A,B) concatenates A and B?

**L314**: Can you confirm that the unit matrix here is different from the identity matrix?

**Eq. (20)**: Shouldn't \tilde z have a subscript B?

**Fig. 4**: It will be good to include in-text interpretation of these results.

**Fig. 6** and text describing it (**L344-L349**): Please have a more detailed explanation of these steps.

**L353**: Write down the learning rate value explicitly.

**L356**: You mean the simulation distribution of Braverman et al. (2021)?

**L358-L360**: Please expand the discussion here, there is lots of information in Figs. 7 and 8 that is left uncommented.

**L366-L372**: Remind the readers of the practical significance for having good approximations of the averaging kernel A.

**Table 2**: Did you compute the Jacobian for the ReFRACtor model analytically or numerically? It will be insightful to know how challenging it is to compute the derivatives of the physical model.

**L398**: I believe the abbreviation QoI has not been defined before. Do you mean OI?

**Eq. (25)**: Are the random variables epsilon and delta independent from each other? If so, one can perhaps write Eq. (25) more compactly with a single noise term that combines the mean and covariances of epsilon and delta.

**L440**: A reference on the 1% relative error in operator learning frameworks will be helpful.

**Replies**

We'd like to thank the reviewer for great comments and some constructive criticism. The observations and suggestions in the review will surely improve the quality and readability of the article. Please find below replies to specific comments:

GENERAL COMMENTS

1. **Methodology**: The comment about technicality is appreciated, but we will have to add that a large part of why the method works so well is due to technical improvements we have introduced. Depending on editorial choices, more in-depth discussion could be moved to an appendix if it hurts the general flow of the text, but this would in a sense contradict what the second reviewer asked which is more technical details. We will include the reviewers suggestions on describing the computational virtues of GP approach via speed and interpretability increases compared to some competing methods. On the "flow" in Kernel Flows, the name comes from the original publication by Owhadi et al. and has more to do with the non-parametric version therein. The novel cross-validation approach draws inspiration from the original work with mini-batching and simple gradient descent as opposed to the commonly used Maximum Likelihood Estimation used to train GP's. We will highlight these differences better in the revised manuscript.

2. **Technical comments on the GP presentation:** We appreciate the noted similarities to OI/OE which are generally more familiar to the target audience of this publication. OI/OE use Gaussian assumptions to derive a mean and covariance for the posterior distribution as a solution to an inverse problem, i.e. data assimilation or a retrieval. In Gaussian Process Regression, the target function (here, the forward model) is represented similarly by a Gaussian distribution that has a mean (prediction) and covariance (error estimate of the prediction). Similarities in formulas come from using Gaussian distributions in both cases. Including a short remark in the end of GP discussion could indeed improve the readability and intuition behind using GPs, albeit using it to solve a different problem.

3. **Figure captions and in-text figure interpretations**: We take note of the comment on figure caption descriptiveness and will make effort to add additional information to both figure descriptions and in-text discussions when necessary.

SPECIFIC COMMENTS

**L28**: We will add a specific reference on this.

**L31-L32**: Total-colum mole fraction denotes the average amount of CO2 over a vertical column of air in a specific ground pixel, generally meaning a pressure weighted average over concentrations on different altitudes. We will make this more clear in-text.

**L46**: We will consult the Algorithm team to get specific numbers and statistics.

**L68**: Model atmospheres here refer to outputs of computational atmospheric models, like the Copernicus Atmospheric Monitoring Services (CAMS) model used in the referenced publications. We will add a description.

**L90**: The distinction between y and z is deliberate: y is a physical quantity, while z is a principal component loading

**Eq. (2)**: we refer here specifically to the posterior of the GP, not the posterior of solving the inverse problem in question. This distinction might be a bit confusing and we will make it more clear that we mean the GP, not the retrieval.

**L117**: the kernel function in question is explicitly written out in equation (3). We will make a reference to it.

**L119-L120**: we will move the explanation earlier to make the text flow better.

**L125**: we will implement the proposed change.

**Eq. (5)**: we will add a reference to e.g. Rasmussen: MLE is standard way of optimizing kernel parameters and deriving it would be out of scope for this publication.

**L145**: as these two concepts are more or less exchangeable, we will refer to RMSE instead.

**Eq. (6)**: For clarity, the first term in the equation is the same as in eq. (1), that is the covariance function evaluated at a new point x* against all the training data, here limited to be withing the mini-batch but leaving out the i:th data point. The next term, in paranthesis, is a rank 1 down-date to the full data covariance matrix also present in eq 1, obtained by the formulas following Stewart (1998) and Zhu et al. (2022) as stated in our text. We will make this even more explicit and easier to follow in revision.

**L156**: a regular Euclidean norm is rising the vector in question to 2:nd power and taking a sum over the elements. A k-norm means simply by replacing the 2:nd power by k:th power, where k can be any integer. We will likely omit this since in general we do use the 2:nd power and it provides enough regularization for the method to work.

**L160-L166**: Taking closed form derivatives could speed up the computation here. We used automatic differentiation in learning the kernel parameters simply out of convenience, since our mini-batches are small and the dimension-reduced parameters have a low dimensionality, using automatic differentiation doesn't result in a lot of computational burden. One could implement closed form derivatives here as well, but in this case it would be needed to hand-code separately and we haven't implemented that yet. This could be done in a future version of the algorithm to gain even faster training performance. Using automatic differentiation is described here simply due to transparency of algorithm implementation.

**Line 1 of Algorithm 1**: Initialization is simply to allocate a vector of zeros for holding the loss function values. At step 1, the 1$^{st}$ element of R is assigned to contain the loss function values coming from theta_1, so what the reviewer suggests in initialization is actually taking place during step 1 of iteration.

**Section 2.4**: we tested Latin Hypercubes and random sampling to span the space of available parameters. This results usually in "holes" in parameter space, which means that interpolation using a GP can have areas with inconsistent performance. As we are aiming for a model that performs well everywhere, we want an optimal space filling design in high dimensions, which can be achieved using Sobol sequences.

**FP model (L190-L215)**: The confusion is understandable as the forward model F() is not a simple function. We try to illustrate that a part of what happens in the function is the radiative transfer (eq.9), which is then convolved with instrument line shape (eq. 10), and that we are interested in emulating the expensice RT part. Detailed description of the forward model is given in the OCO-2 ATBD, which contains lengthy physics considerations that are out of the scope of this publication. We can make the above points clearer in this section to avoid confusion.

**L307**: As defined in Braverman et al. templates are a geophysical areas at certain time intervals where conditions are more or less similar in all pictures. This helps to constrain the UQ analysis in the source article to statistically similar areas, which is repeated as necessary when conditions change. For our purposes, we are just taking one realistic geophysical location and demonstrating the method on it. We will remove the reference to templates to avoid confusion and refer to a set of realistic geophysical conditions from real world measurements.

**Eq. (19)**: diag() is defined to be a block-diagonal matrix, that is, each input is a block in the block diagonal structure, while all other entries are zeros. We will consider a more standard notation alternative here.

**L314**: we will refer to indentity matrix instead of unit matrix.

**Eq. (20):** for consistency with Figure 6, we will add a subscript _B

**Fig. 4**: we will add more interpretation in-text. In short the figure is included to show the kinds of features in the spectra that these principal components encode, which can be

visually informative for readers with more spectroscopy background. At the same time, these principal components are not physical and might contain features not directly related to any one physical phenomenon.

**Fig. 6**: we will consider adding more descriptive narrative on what happens in the flow diagram, and refer to appropriate equations introduced earlier. The flow chart is meant as a conclusion on the section, to repeat how the entire architecture works in higher level.

**L356**: Yes: we will clarify that we use the same distribution as Braverman et al.

**L358-L360**: we will expand the discussion. Information described is generally how well the training works: Fig (7) shows the loss function converging to a small value, which demonstrates the training finding a stable minimum value after initial oscillations. Second panel shows the evolution of different kernel parameters and that a stable state is achieved during training, or that the parameter values don't change significantly anymore towards the end. The last panel of figure (7) demonstrates true vs predicted label values being really close to 1-to-1 line. Same story is illustrated in figure (8), where for each label, or principal component loading, we can see that finding optimal kernel parameters yields very accurate predictive performance over the test set. These labels will then be assembled into radiances accoring to the flow chart in figure (6). We will elaborate on these points in the revised manuscript.

**Table 2**: Numerically; the full physics forward model includes a built-in way of computing derivatives that are operationally used in OE retrievals.

**L398**: QoI denotes Quantity of Interest, which in our case is the column-averaged CO2, or XCO2. we will spell this out in revision.

**Eq. (25)**: they are independent: epsilon comes from operational noise model, while delta is model discrepancy describing model-real physics mismatch. These should for this reason be written out explicitly.

**L440**: we will reference operator learning literature for this claim.

**Reviewer 2**

The Author present a machine learning based radiative transfer emulator to speed up OCO CO2 retrievals. The method is based on a novel formulation known as Gaussian process regression, and also provides uncertainty estimates alongside simple regression outputs. The emulator is evaluated against independent radiative transfer simulations and is also used in CO2 retrievals from synthetic spectra, revealing a performance similar to that of the full algorithm although somewhat different retrieval uncertainties.

The presented methodology appears sound, even though I found section 2 hard to follow at times, as a number of highly technical concepts (Matérn kernel, Sobol sequence, reproducing Hilbert space norm) are introduced without much explanation for what they are. I would invite the Authors to clarify these concepts better in the revised version to make the paper more readable.

MAIN COMMENTS

- P4, eq. 1. I find the notation confusing. It is not clear to me what the function Gamma operates on. Does it operate on a vector and a matrix (as in Gamma(x*, X)) or on two matrices (as in Gamma(X,X))? Consider using a clearer notation.

- P4, eq. 2. What relationship exists between Gamma(x*, X) and Gamma(X, x*)? Are they the same? Are they one the transpose of the other?

- P4, L110-111. "... sets GP regression apart from many neural network based machine learning models". Given that models such as, e.g., multilayer perceptron and radial basis function networks are also analytically differentiable, input uncertainty can be linearly propagated through them. In this sense, what is it that can be done with GR but not with MLP or RBF?

- P4, L117. The concept of Matérn kernel is invoked but it is not defined. What is such kernel and why is it relevant here?

- P6, L139. It is taken for granted that the reader knows what a "relative reproducing kernel Hilbert space norm" is, but I am quite sure this is not the case for many.

- P6-7, L169-170. What is a Sobol sequence and why is it relevant in this context? I think it is too much to expect the reader to go through the Sobol (1967) without at least a summary of what this all is about. Furthermore, as far as I have seen, this paper is only available in Russian, so I would have had a hard time going through it even if I decided to embark on the endeavour while writing my review.

- P14, L316. Again, the reader is referred to section 2.4 for the concept of Sobol sequence, but also there the concept is just mentioned, and not much is said of how this method works.

MINOR COMMENTS

- P1, L10. Isn't there a more recent report?

- P2, L32. Also add the Chinese TanSat (Ran and Li, 2019, doi: 10.1016/j.scib.2019.01.019).

- P2, L36. Also add TanSat-2 (Wu et al., 2023, doi: 10.3390/rs15204904). Furthermore, I think the GeoCarb mission was cancelled towards the end of 2022.

- P3, L69. "to" -> "for", "with"?

- P4, L110. "to preclude prediction uncertainties". Do you mean "to predict"?

- P7, eq. 9. Notation is rather uncommon. Usually zenith angles are denoted by thetas instead of taus (you also do that in subsequent sections), whereas tau is reserved for optical depths, which, I think, you denote by g(lambda) (at least, if I correctly interpret your equation g(lambda) seems to be the slant optical depth, you may want to mention that)

- P8, L205. "to" -> "on"

- P8, L206. In principle fluorescence is additional radiation coming from the surface, it is not an effect of the observing system. I see later on that in your retrieval it is treated as an instrumental offset, but it may be best to keep the concepts separate at this point of the manuscript.

- P13, L299, "wave length" -> "wavelength"

- P13, L318, remove "of" between "simplifies" and "computations"

- P31, Owhadi and Yoo (2018) and Owhadi and Yoo (2019) seem to be the same paper. Either it is a duplicated entry or one of the two entries is incorrect.

**Replies**

We want to thank the reviewer for a thorough review and constructive criticism and comments provided. These are sure to make the article a more coherent publication.

The issue identified with technical concepts will be remedied in the revised manuscript and we will provide further details for improving readability of the article. In short, Matérn kernel is a more expressive choice of kernel compared to the usual Gaussian / radial basis functions used by default in Gaussian Process Regression, which tend to be "too smooth" to capture more abrupt changes in the function that is being approximated. Sobol sequences are a space filling desing akin to Latin Hypercubes, providing optimality (observed experimentally) in generation of training data. Reproducing Kernel Hilbert space norm is a way, in high level, to measure the smoothness of a function approximation achieved with kernel methods.

Reply to Main Comments, in order:

-- P4, eq. 1: Gamma is shorthand for operating on both vectors and matrices, and denotes a covariance matrix. We will clarify the notation by explicitly writing out what is meant by Gamma(x*, X)  and Gamma(X,X).

- P4, eq. 2: Gamma(x*, X) and Gamma(X, x*) are transposes of each other. We will clarify this in the revised manuscript.

- P4, L110-111: Linear propagation of uncertainty is indeed possible using e.g. MLP or RBF. GP regression is different since it provides, as an output, a distribution with mean being the predicted value, and (co-)variance the calibrated uncertainty. When calibrated correctly, this makes the GP directly self-aware of its own prediction uncertainty. This is added on top of linear error propagation provided by differentiable models, and can be added to the total error budget e.g. by modifying the likelihood function to include prediction uncertainty in addition to measurement error. We will expand on this comment in the revised version.

- P4, L117: as stated above, we will clarify the benefits of using a Matérn kernel as opposed to more standard choices.

- P6, L139: as stated above, we will clarify the meaning of RKHS norm and why it is relevant.

- P6-7, L169-170: we will add further references to Sobol Sequences. A more recent description, in english, can be found from e.g. the Numerical Recipes books (https://iate.oac.uncor.edu/~mario/materia/nr/numrec/f7-7.pdf)

Reply to Minor Comments, in order:

- P1, L10: there is indeed a more recent report, we will reference accordingly in the revised version.

- P2, L32: thank you for the suggestion, we will add mention of TanSat

- P2, L36: we will add mention of TanSat2 and omit GeoCarb, as suggested.

- P7, eq. 9: we will change the mentions to tau to refer to thetas instead to more accurately comply with standard notation in the field.

- P8, L206: fluorescence is part of the forward model F(x) we are trying to emulate, but the reviewer is correct in pointing out that it's not part of the instrument effects, although it's applied after the radiative transfer calculations which are our main concern in emulation. We will clarify this distinction as suggested and mention that SIF is added after the fact as additional radiation coming from the surface.

---

## Author Response (AR2)

**Author's Response**

-Otto M. Lamminpää

Response to additional comments by the Editor:
* * *
1) In reference to my general comment 1, the first paragraph of Section 2 discusses the potential to enforce physical constraints, such as parameter bounds, during the GP training process. This is an important aspect, and I suggest the authors include a few more sentences explaining how this issue is handled within their retrieval application.

2) In their response letter, the authors acknowledge the mathematical parallels between GPs and the OE/OI framework (e.g., both employing Gaussian assumptions). However, this discussion does not appear in the updated text.

3) The authors note, "L46: We will consult the Algorithm team to get specific numbers and statistics", but they did not include this information in the revised manuscript.
* * *
Reply:

1) Added discussion and clarification on how the training data spans the intervals containing the maximum and minimum values of each input parameter, and that withing those limits GP regression can be seen as interpolation. Furthermore, if the input point is outside that interval, prediction uncertainty is large by design.
2) We have added a remark in section 2.1. outlining the similarities and differences between OI/OE and GPR.
3) We have added a reference and a sentence stating that approximately 1/3 of measurements are currently processed.